# Sensitivity of the Antarctic ice sheets to the warming of Marine Isotope Substage 11c

Martim Mas e Braga[1,2], Jorge Bernales[3], Matthias Prange[3], Arjen P. Stroeven[1,2], and Irina Rogozhina[3,4]

[1]Geomorphology & Glaciology, Department of Physical Geography, Stockholm University, Stockholm, Sweden
[2]Bolin Centre for Climate Research, Stockholm University, Stockholm, Sweden
[3]MARUM - Center for Marine Environmental Sciences, University of Bremen, Bremen, Germany
[4]Department of Geography, Norwegian University of Science and Technology, Trondheim, Norway

**Correspondence:** Martim Mas e Braga (martim.braga@natgeo.su.se)

**Abstract.** Studying the response of the Antarctic ice sheets during periods when climate conditions were similar to the present can provide important insights into current observed changes and help identify natural drivers of ice sheet retreat. In this context, the Marine Isotope Substage 11c (MIS11c) interglacial offers a suitable scenario, given that during its later portion, orbital parameters were close to our current interglacial. Ice core data indicate that warmer-than-present temperatures lasted for longer than during other interglacials. However, the response of the Antarctic ice sheets and their contribution to sea level rise remain unclear. We explore the dynamics of the Antarctic ice sheets during this period using a numerical ice-sheet model forced by MIS11c climate conditions derived from climate model outputs scaled by three glaciological and one sedimentary proxy records of ice volume. Our results indicate that the East and West Antarctic ice sheets contributed 4.0–8.2 m to the MIS11c sea level rise. In the case of a West Antarctic Ice Sheet collapse, which is the most probable scenario according to far-field sea level reconstructions, the range is reduced to 6.7–8.2 m independently of the choices of external sea-level forcing and millennial-scale climate variability. Within this latter range, the main source of uncertainty arises from the sensitivity of the East Antarctic Ice Sheet to a choice of initial ice sheet configuration. We found that the warmer regional climate signal captured by Antarctic ice cores during peak MIS11c is crucial to reproduce the contribution expected from Antarctica during the recorded global sea level highstand. This climate signal translates to a modest threshold of 0.4 °C oceanic warming at intermediate depths, which leads to a collapse of the West Antarctic Ice Sheet if sustained for at least 4 thousand years.

## 1 Introduction

Lasting for as much as 30 thousand years (kyr), between 425 and 395 thousand years ago (ka), Marine Isotope Substage 11c (hereafter MIS11c) was the longest interglacial of the Quaternary (Lisiecki and Raymo, 2005; Tzedakis et al., 2012). It also marked the transition from weaker to more pronounced glacial-interglacial cycles (EPICA Community Members, 2004). Its long duration is attributed to a modulation of the precession cycle, resulting in $CO_2$ levels that were high enough to suppress the cooling of the climate system due to the low eccentricity and thus reduced insolation (Hodell et al., 2000). Moreover, ocean sediment cores (e.g., Hodell et al., 2000) and climate models (e.g., Rachmayani et al., 2017) show that the MIS11c global overturning circulation was at an enhanced state, resulting in asynchronous warming of the southern and northern high

latitudes (i.e., they did not reach their warming peak at the same time; Steig and Alley, 2002). However, Dutton et al. (2015)
point out that climate modelling experiments with realistic orbital and greenhouse gas forcings fail to fully capture this MIS11c warming despite the fact that orbital parameters were almost identical to Present Day (PD) during its late stage (cf. EPICA Community Members, 2004; Raynaud et al., 2005). Earlier studies (e.g., Milker et al., 2013; Kleinen et al., 2014) have shown that climate models also tend to underestimate climate variations during MIS11c, for which ice core reconstructions show the mean annual atmospheric temperature over Antarctica to have been about 2 °C warmer than Pre-Industrial (PI) values.

A better understanding of the climate dynamics during Quaternary interglacials, especially those that were warmer than today, is critical because they can help assess Earth's natural response to future environmental conditions (Capron et al., 2019). Among these periods, MIS 5e (also referred to as the Eemian, Last Interglacial, or LIG; Shackleton et al., 2003) was originally proposed to be a possible analogue for the future of our current interglacial (Kukla, 1997). More recently, MIS11c has been considered another suitable candidate, since its orbital conditions were closest to PD (Berger and Loutre, 2003; Loutre and Berger, 2003; Raynaud et al., 2005). Furthermore, ice core evidence indicates that Termination V (i.e., the deglaciation that preceded MIS11) was quite similar to the last deglaciation in terms of rates of change in temperature and greenhouse gas concentrations (EPICA Community Members, 2004). The unusual length of MIS11c and a transition to stronger glacial-interglacial cycles seen in the subsequent geological record may have been triggered by a reduced stability of the West Antarctic Ice Sheet (WAIS, Fig. 1). The latter may have been due to the cumulative effects of the ice sheet lowering its bed (Holden et al., 2011), which in turn provided a positive climate feedback (Holden et al., 2010). The long duration of MIS11c was also shown to be a key condition to triggering the massive retreat of the Greenland Ice Sheet (GIS; Robinson et al., 2017). Elucidating the response of the Antarctic ice sheets (AIS) to past interglacials can also help identify various triggers of ice sheet retreat. This is because each interglacial has its unique characteristics: for example, while MIS11c was longer than the LIG, the latter was significantly warmer (Lisiecki and Raymo, 2005; Dutton et al., 2015).

The MIS11c history of Antarctica is less constrained than that of Greenland (e.g., Willerslev et al., 2007; Reyes et al., 2014; Dutton et al., 2015; Robinson et al., 2017). Whereas Raymo and Mitrovica (2012) consider that the WAIS had collapsed and that the East Antarctic Ice Sheet (EAIS, Fig. 1) provided a minor contribution based on their estimate of MIS11c global sea levels of 6 to 13 m above PD, studies directly assessing the AIS response have been elusive. For example, sedimentary evidence has been inconclusive regarding the possibility of a collapse of the WAIS during some Quaternary interglacials (Hillenbrand et al., 2002, 2009; Scherer, 2003), and evidence for the instability of marine sectors of the EAIS has only recently been provided (Wilson et al., 2018; Blackburn et al., 2020). Counter-intuitively, the dating of onshore moraines in the Dry Valleys to MIS11c, indicating local ice advance, has been used to indirectly support regional ice sheet retreat (Swanger et al., 2017). Swanger et al. (2017) argue that ice sheet retreat in the Ross Embayment provided nearby open-water conditions and therefore a source of moisture and enhanced precipitation, fueling local glacier growth. Previous numerical modelling experiments that encompass MIS11c also lack a consensus regarding AIS volume changes. For example, Sutter et al. (2019) report an increased ice volume variability from MIS11 (i.e., the isotopic stage in which MIS11c lies) onwards, caused by stronger atmospheric and oceanic temperature variations, while Tigchelaar et al. (2018) only obtained significant volume changes during the last 800 kyr when increasing their ocean temperatures to values as high as 4 °C. Conversely, de Boer et al. (2013) report higher sea level

contributions during MIS 15e, 13, and 9, and weaker contributions during MIS 11c and 5e. Among the past interglacials, the LIG and Pliocene are considered to be the closest analogues to MIS11c, and studies acknowledge the possibility of a WAIS collapse in both periods (e.g., Hearty et al., 2007; Naish et al., 2009; Pollard and DeConto, 2009). However, Pliocene model results were shown to be highly dependent on the choice of climate and ice-sheet models (de Boer et al., 2015; Dolan et al., 2018).

Constraints are also scarce for the MIS11c climate, and its heterogeneity is reflected in the ice core records. Reconstructions from different ice cores located in East Antarctica (circles in Fig. 1) show different histories regarding the evolution of atmospheric surface temperature. For example, the Vostok ice core surface air temperature reconstruction (Petit et al., 1999; Bazin et al., 2013) reveals a weak temperature peak (about 1.6 °C above PI around 410 ka) compared to those of EPICA Dome C (EDC; over 2.7 °C above PI around 406 ka,  Jouzel et al., 2007) and Dome Fuji (DF; 2.5 °C above PI around 407 ka, Uemura et al., 2018). The latter two ice-core records also present a peak-warming period of much longer duration (ca. 15 kyr compared to 7 kyr at Vostok).

As detailed, many modelling studies have investigated AIS responses over time periods that include MIS11. However, so far none has focused specifically on this period. Given the scarce information for MIS11c and conflicting constraints on how Antarctica responded to this exceptionally long interglacial (Milker et al., 2013; Dutton et al., 2015), we here focus on MIS11c, the peak warming period between 420 and 394 ka. Our aim is to reduce the current uncertainties in the AIS behaviour during MIS11c, addressing the following questions:

1. How did the AIS respond to the warming of MIS11c? More specifically, what are the uncertainties in the AIS minimum configuration, timing and potential sea level contribution?

2. What was the main driver of the changes in the AIS volume? Was it warming duration, peak temperature, changes in precipitation, or changes in the oceanic forcing?

Ice-sheet model simulations depend on applied forcings, boundary conditions, and parameterisations for a wide range of processes. Such parameters control, for example, basal sliding, ice deformation, bedrock deformation, ice-shelf basal melting, and ice-shelf calving. The sensitivity of ice volume changes across glacial-interglacial time scales to model parameters was extensively explored by Albrecht et al. (2020). DeConto and Pollard (2016) carried out a large ensemble analysis for the LIG and the Pliocene, where parameters related to ice-shelf loss were constrained according to their ability to simulate target ranges of sea-level contribution. Simpler flow-line models have also been used to evaluate uncertainties in basal conditions (Gladstone et al., 2017) and flow-law parameters (Zeitz et al., 2020). Here, we perform five ensembles of experiments that focus on choices that are external to the numerical model, and could help guide other modelling efforts on the choice of forcings and boundary conditions. We evaluate the impact of the following on AIS volume and extent during MIS11c: the choice of proxy record (including their differences in signal intensity and structure), the choice of sea level reconstruction, and uncertainties in assumptions regarding the geometry of the AIS at the start of MIS11c.

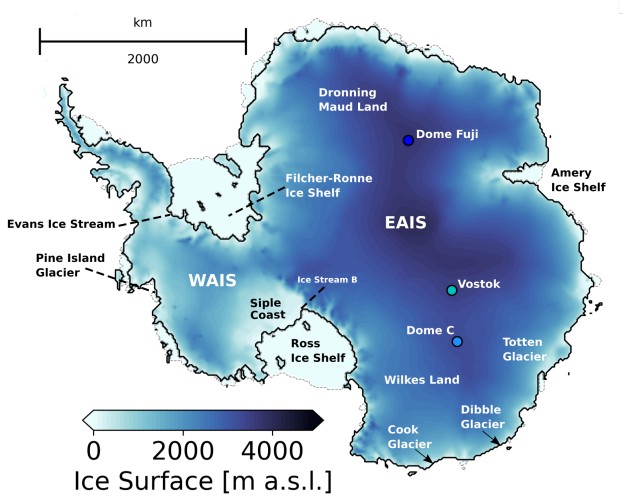

**Figure 1.** Surface topography of the AIS at the start of our core experiments (425 ka), based on a calibration against Bedmap2 (Fretwell et al., 2013, , see Sect. 2.1). Locations mentioned in the text are showcased, including the drilling sites of the ice cores used in this study (circles).

## 2 Methods

### 2.1 Ice-sheet model

For our experiments we employ the 3D thermomechanical polythermal ice-sheet model SICOPOLIS (Greve, 1997; Sato and Greve, 2012) with a 20 km horizontal grid resolution and 81 terrain-following vertical layers. It uses the one-layer enthalpy scheme of Greve and Blatter (2016), which is able to correctly track the position of the cold-temperate transition in the thermal structure of a polythermal ice body.

The model combines the Shallow Ice Approximation (SIA) and Shelfy Stream Approximation (SStA) using (c.f. Bernales et al., 2017b, Eq. 1)

$$\mathbf{U} = (1 - w) \cdot \mathbf{u}_{\mathrm{sia}} + \mathbf{u}_{\mathrm{ssta}}, \tag{1}$$

where $\mathbf{U}$ is the resulting hybrid velocity, $\mathbf{u}_{\mathrm{sia}}$ and $\mathbf{u}_{\mathrm{ssta}}$ are the SIA and SStA horizontal velocities, respectively, and $w$ is a weight computed as

$$w(|\mathbf{u}_{\mathrm{ssta}}|) = \frac{2}{\pi} \arctan\left( \frac{|\mathbf{u}_{\mathrm{ssta}}|^2}{u_{\mathrm{ref}}^2} \right), \tag{2}$$

where the reference velocity $u_{\mathrm{ref}}$ is set to 30 ma$^{-1}$, marking the transition between slow and fast ice. This hybrid scheme reduces the contribution from SIA velocities mostly in coastal areas of fast ice flow and heterogeneous topography, where this approximation becomes invalid. Basal sliding is implemented within the computation of SStA velocities as a Weertman-type

law (cf. Bernales et al., 2017a, Eqs. 2–6). The amount of sliding is controlled by a temporally fixed, spatially varying map of friction coefficients that was iteratively adjusted during an initial present-day equilibrium run (cf. Pollard and DeConto, 2012b), such that the grounded ice thickness matches the present-day observations from Bedmap2 (Fretwell et al., 2013) as close as possible. Sliding coefficients in sub-ice shelf and ocean areas are set to $10^5 \, \mathrm{m\,a^{-1}\,Pa^{-1}}$, representing soft, deformable sediment, in case the grounded ice advances over this region. The initial bedrock, ice base, and ocean floor elevations are also taken from Bedmap2. Enhancement factors for both grounded and floating ice are set to 1, based on sensitivity tests in Bernales et al. (2017b). This choice provides the best match between observed and modelled ice thickness for this hybrid scheme, similar to the findings in Pollard and DeConto (2012a).

Surface mass balance is calculated as the difference between accumulation and surface melting. The latter is computed using a semi-analytical solution of the positive degree day (PDD) model following Calov and Greve (2005). Near-surface air temperatures entering the PDD scheme are adjusted through a lapse rate correction of $8.0 \, °\mathrm{C \, km^{-1}}$ to account for differences between the modelled ice sheet topography and that used in the climate model from which the air temperatures are taken. For the basal mass balance of ice shelves, we use a calibration scheme of basal melting rates developed in Bernales et al. (2017b) to optimise a parameterisation based on Beckmann and Goosse (2003) and Martin et al. (2011) that assumes a quadratic dependence on ocean thermal forcing (Holland et al., 2008; Pollard and DeConto, 2012a; Favier et al., 2019). This optimised parameterisation is able to respond to variations in the applied Glacial Index (GI, Sect. 2.2) forcing. A more detailed description of this parameterisation is given in Sect. 1 of the supplementary material. In our experiments, we prescribe a time lag of 300 years for the ocean response to GI variations, which is considered the most likely lag in response time of the ocean compared to the atmosphere in the Southern Ocean (Yang and Zhu, 2011). At the grounding line, the basal mass balance of partially floating grid cells is computed as the average melting of the surrounding, fully floating cells, multiplied by a factor between 0 and 1 that depends on the fraction of the cell that is floating. This fraction is computed using an estimate of the sub-grid grounding line position based on an interpolation of the current, modelled bedrock and ice-shelf basal topographies. At the ice shelf fronts, calving events are parameterised through a simple thickness threshold, where ice thinner than 50 m is instantly calved away.

Bed deformation is implemented using a simple elastic lithosphere, relaxing asthenosphere (ELRA) model, with a time lag of 1 kyr and flexural rigidity of $2.0 \times 10^{25} \, \mathrm{N\,m}$, which Konrad et al. (2014) found to best reproduce the results of a fully-coupled ice sheet–self-gravitating viscoelastic solid Earth model. The geothermal heat flux applied at the base of the lithosphere is taken from Maule et al. (2005) and is kept constant. All relevant parameters used in the modelling experiments are listed in Table 1.

Sea-level contribution at a given time step is computed in SICOPOLIS as the difference in total ice volume above flotation between the ice sheet at the time step and the spun-up Pre-Industrial ice sheet. When computing ice volume, differences in bedrock elevation between the two ice sheets are accounted for by using a common reference bedrock elevation in all time steps. We also correct for the projection effect on the horizontal grid area.

All ensembles cover a period from 420 to 394 ka. After the calibration for basal sliding mentioned above, we initialise the AIS by performing a thermal spin-up over a period of 195 kyr from 620 to 425 ka, i.e., apply a transient surface temperature signal from the EDC ice core (Jouzel et al., 2007) as an anomaly to our PI climate (described in the next section) while keeping

**Table 1.** Main parameters used in the experiments.

| Parameter | Name | Value | Units |
|---|---|---|---|
| $E_{grounded}$ | Enhancement factor (grounded ice) | 1 | |
| $E_{floating}$ | Enhancement factor (ice shelves) | 1 | |
| n | Glen's Flow Law exponent | 3 | |
| $p$ | Weertman's Law p exponent | 3 | |
| $q$ | Weertman's Law q exponent | 2 | |
| $\tau$ | ELRA model time lag | 1 | kyr |
| $D$ | ELRA model flexural rigidity | $2.0 \times 10^{25}$ | Nm |
| $\gamma_{lr}$ | Lapse rate correction | 8.0 | $°C\,km^{-1}$ |
| $S_0$ | Sea water salinity | 35 | |
| $\rho_{sw}$ | Sea water density | 1028 | $kg\,m^{-3}$ |
| $\rho_{ice}$ | Ice density | 910 | $kg\,m^{-3}$ |
| $c_{p0}$ | Ocean mixed layer specific heat capacity | 3974 | $J\,kg^{-1}\,K^{-1}$ |
| $\gamma_T$ | Thermal change velocity | $10^{-4}$ | $m\,s^{-1}$ |
| $L_i$ | Latent heat of fusion | $3.35 \times 10^5$ | $J\,kg^{-1}\,K^{-1}$ |

the ice sheet geometry constant at our previously calibrated Bedmap2-based configuration. We then let the AIS freely evolve for 5 kyr, between 425 and 420 ka, applying transient GI forcing during the relaxation period (Fig. S12). We chose 425 ka as the starting point for relaxation because it is when the MIS11c oxygen isotope values in the EDC ice core are closest to PI. When analysing the results, we ignore the first 5 kyr (425–420 ka) to avoid a shock from suddenly letting the ice-sheet topography freely evolve at the start of our period of interest. Figure 1 shows the thermally spun-up ice sheet configuration at 425 ka, from which the simulations start. The EDC ice core was chosen for the thermal spin-up and as common forcing for all ensemble runs except for CFEN, where we test different core-derived climate signals (see below), because it spans the longest period among the three ice cores tested, while still providing a relatively high temporal resolution.

## 2.2 Climate forcing and core experiments

In an effort to assess the impact of similarities and differences in existing paleoclimate reconstructions, and regional differences in the ice-core records, we perform an ensemble of simulations where each member is forced by a GI (Eq. 3) derived from $\delta D$ from ice cores, or $\delta^{18}O$ from the LR04 stack of deep-sea sediment cores (Fig. 2a; Petit et al., 2001; EPICA Community Members, 2004; Lisiecki and Raymo, 2005; Uemura et al., 2018). Since an ensemble of fully coupled climate-ice sheet model runs over 26 kyr is at present computationally challenging, an evaluation of possible scenarios for the peak-temperature response during MIS11c based on the paleoclimate signals from different ice sheet sectors can be a cheaper, yet effective approach. The GI method is a way of weighting the contributions from interglacial (PI) and full glacial (Last Glacial Maximum; LGM)

**Table 2.** Ice and sediment cores reference values used in Eq. (3), together with the age (in thousand years before present; ka) from which the LGM reference values were obtained. The respective age models of each core, and their references, are listed.

| Record | Type (isotope) | $\delta X_{PI}$ [‰] | $\delta X_{LGM}$ [‰] | Age (ka) | Age model | Reference |
|--------|----------------|--------------------|--------------------|----------|-----------|-----------|
| EDC | Ice ($\delta$D) | -397.4 | -449.3 | 24.0 | EDC3 | EPICA Community Members (2004) |
| DF | Ice ($\delta$D) | -425.3 | -469.5 | 22.8 | AICC2012 | Uemura et al. (2018) |
| Vostok | Ice ($\delta$D) | -440.9 | -488.3 | 24.4 | GT4 | Petit et al. (2001) |
| LR04 | Sediment ($\delta^{18}$O) | 3.23 | 4.99 | 20.0 | LR04 | Lisiecki and Raymo (2005) |

average states. It does so by rescaling a variable curve (usually temperature or isotope reconstructions from an ice or sediment record) based on reference PI and LGM values, which consider PI climate as GI = 0 and LGM climate as GI = 1 (Eq. 3):

$$GI(t) = \frac{\delta X(t) - \delta X_{PI}}{\delta X_{LGM} - \delta X_{PI}} \tag{3}$$

Where $t$ is time, and X is deuterium for the ice cores or $^{18}$O for sediment cores. The value for $\delta X_{PI}$ was obtained as the average of the last 1000 years before 1850 CE, while $\delta X_{LGM}$ was taken as the minimum and maximum value for $\delta$D and $\delta^{18}$O, respectively, between 19 and 26.5 ka (cf. Clark et al., 2009; Clason et al., 2014). For our two reference climate states (i.e., PI and LGM), we use the Community Climate System Model version 3 (CCSM3) PI time slice in Rachmayani et al. (2016), and the LGM time slice in Handiani et al. (2013), which used identical model versions and were run on the same

platform. A brief assessment of the model biases against PD data is provided (Sects. 2 and 3 of the supplementary material). The atmospheric and ocean temperature (T) fields at time $t$ are reconstructed based on their respective PI and LGM reference fields ($T_{PI}$ and $T_{LGM}$ respectively) using (see also Fig. S13):

$$T(t) = T_{PI} + GI(t) \cdot (T_{LGM} - T_{PI}) \tag{4}$$

while precipitation is given by an exponential function to prevent negative values and to ensure a smooth transition between

the PI and LGM states:

$$P(t) = P_{PI}{}^{1-GI(t)} \cdot P_{LGM}{}^{GI(t)} \tag{5}$$

The PI and LGM reference values (including the reference ages for the latter) for the three ice cores and the LR04 stack are summarised in Table 2, together with their respective age models. The ensemble of simulations forced by different GI curves (Climate Forcing ENsemble, CFEN) constitutes our core experiments.

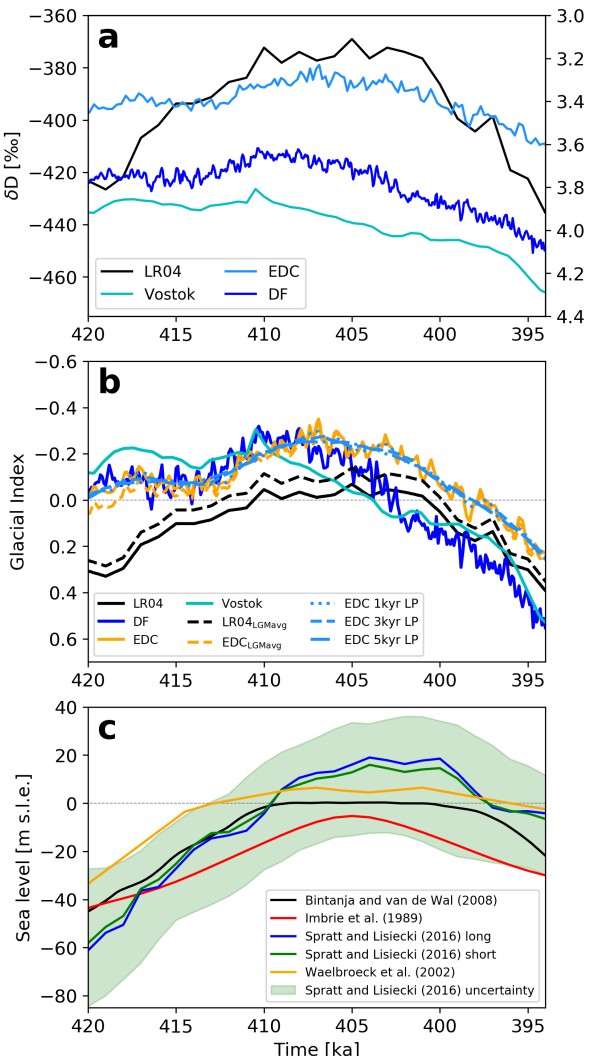

**Figure 2.** Reconstructions used in this study: (a) LR04 $\delta^{18}$O (black) and Vostok, Dome C (EDC), and Dome Fuji (DF) ice-core $\delta$D [‰]; (b) resulting Glacial Indices from the reconstructions in (a) (cf. Sect. 2 and Table 3 for the legends); (c) global mean sea level anomaly relative to PI (meter sea level equivalent, m s.l.e.).

## 2.3 Sensitivity experiments

### 2.3.1 Sensitivity to the GI scaling

Because different approaches have been used to transform the isotope curves into a GI, we assess the sensitivity to the choice of the scaling procedure by performing an additional scaling using another reference value for $\delta X_{LGM}$. In the new scaling procedure, $\delta X_{LGM}$ is the average (between 19 ka and 26.5 ka) rather than the peak value. We compare the effects of using

these two procedures when applied to the EDC ice core $\delta$D and the LR04 stack $\delta^{18}$O records (orange and black dashed lines in Fig. 2b respectively). We call this ensemble the Scaling Sensitivity ENsemble (SSEN)."

### 2.3.2  Sensitivity to millennial-scale variability

Given the different temporal resolutions of climate records, lower-resolution reconstructions such as LR04 and Vostok might not capture the impact of millennial variability or shorter events, as do EDC and DF (Fig. 2a). Thus, we assess the potential
effects of record data resolution and millennial (or shorter) time scale variability by applying 1, 3, and 5 kyr low-pass filters to the EDC ice core GI and forcing our model with the resulting smoothed GI curves (light blue lines in Fig. 2b). We then compare these three simulations to the original EDC-derived ice sheet history, and call this ensemble the Resolution Sensitivity Ensemble (RSEN).

### 2.3.3  Sensitivity to sea level

Sea level plays an important role in determining the flotation of the ice sheet and the stresses at its marine margins. Uncertainties in global mean sea level reconstructions are therefore a significant concern, and several studies have indeed focused on improving their estimates (e.g., Imbrie et al., 1989; Waelbroeck et al., 2002; Bintanja and van de Wal, 2008; Spratt and Lisiecki, 2016, Fig. 2c). We evaluate the effect of using a particular sea level reconstruction on the evolution of the AIS by running an ensemble of simulations with EDC-derived GI, where each member uses a different sea level reconstruction. For
each ensemble member, the sea level forcing applied at the boundaries of the ice sheet is approximated to the global mean sea level of its respective sea level reconstruction. Sea level curves included in this ensemble are three of the reconstructions presented by Spratt and Lisiecki (2016), termed "long" (i.e., uses records that extend as far back as 798 ka), "short" (uses records that extend at least until 430 ka), and the "upper uncertainty boundary" from their records, because we consider their lower uncertainty boundary to be satisfactorily covered by SPECMAP (Imbrie et al., 1989), which we include. We also include
in the analysis the reconstructions from Bintanja and van de Wal (2008) and from Waelbroeck et al. (2002). All these records are presented in Fig. 2c, and we call this ensemble, where we test different sea level reconstructions, the Sea Level Sensitivity Ensemble (SLSEN).

### 2.3.4  Sensitivity to the choice of initial ice sheet geometry

Similar studies that assess AIS changes over glacial and interglacial cycles often adopt a PI or PD starting geometry (e.g., Sutter
et al., 2019; Tigchelaar et al., 2019; Albrecht et al., 2020). We have followed the same approach in our CFEN experiments (see Sect. 2.2). Although the similarity to the modern AIS configuration has been loosely inferred from sedimentary (Capron et al., 2019) and ice-core (EPICA Community Members, 2004) proxy records, to our knowledge there is no direct evidence to support this claim (e.g., Swanger et al., 2017). Hence, we also perform an ensemble of simulations starting from different ice sheet geometries. This allows for an evaluation of the influence of an initial AIS configuration at 420 ka on its modelled
retreat and advance (including possible thresholds), and provides an uncertainty envelope in its potential sea level contribution

based on this criterion. We call this the Starting Geometry Sensitivity ENsemble (SGSEN), and its three unique geometries are forced with the ice-core reconstructed climate forcings tested in CFEN.

In order to create a representative range of initial geometries at 420 ka, we use a common starting geometry, but vary the relaxation time. For this purpose, we first create an ancillary geometry by perturbing the thermally spun-up AIS with a constant LGM climate (air temperature and precipitation rates) and no sub ice-shelf melting over a 5 kyr period. The resulting ancillary ice sheet (which has an extent that sits between PI and LGM configurations) is then placed at 420, 425 and 430 ka and runs transiently (following the respective GIs) until 394 ka. This creates a representative range of starting geometries at 420 ka (Fig. 3), and each initial ice sheet geometry is labelled gmt1 to gmt3 (Fig. 3a-c; shortest relaxation is gmt1, longest is gmt3). The gmt1 initial topography is generally more extensive and thinner than the control. Its grounding line advanced at the southern margin of the Filcher-Ronne Ice Shelf and at Siple Coast, but the ice sheet interior is on average 200 m thinner than the control and up to 500 m thinner across particular regions such as the dome areas of the WAIS and Wilkes Land (Dome C). It is, however, about 200 m thicker at its fringes, which results in a gentler surface gradient towards the ice sheet margins. The gmt2 initial topography is less than 100 m thinner than the control over the EAIS interior, and about 100 m thicker over the WAIS interior and at the EAIS margins. Finally, the gmt3 initial topography is overall thicker than the control, though not by more than 100 m except at the western side of the Antarctic Peninsula and the WAIS margins, where some regions are up to 300 m thicker (Fig. 3c). Table 3 summarises all experiments described in this section.

## 3 Results

### 3.1 Climate forcing reconstructions

Considering the four adopted isotope curves (Fig. 2a,b), although similar at first sight, the GI reconstructions are different from one another, and therefore offer a range of modelled ice-sheet responses. The LR04 GI reconstruction is generally colder, showing conditions warmer than PI only for the warmest period of MIS11c (i.e., between ca. 410 ka and 400 ka). Consequently, it does not show a peak warming as strong as the other reconstructions (Fig. 2b). Although the ice cores have similar ranges in GI values and similar overall aspects of the curves (and good covariance between EDC and DF; Uemura et al., 2018), they differ in key aspects. The Vostok reconstruction starts at a warmer state than the others at 420 ka, has a modest peak warming at 410 ka, and then consistently declines towards a colder state (crossing the GI = 0 line at about 404 ka). The EDC reconstruction shows a mildly warmer-than-PI state at 420 ka, which persists until about 412 ka. Subsequently, the peak warming starts and persists (in a slightly warmer state than reconstructed with Vostok after 410 ka) until 397 ka. Its rate of decline after 404 ka is similar to the Vostok and LR04 curves, although it is in a warmer state. Finally, the DF reconstruction is somewhere in-between the other two ice cores (Fig. 2b). It shows quite stable conditions at the start (i.e., no pronounced warming), rising to a rather pronounced warming peak similar in structure to the EDC reconstruction, but peaks at 410 ka, similar to the Vostok curve. Finally, its rate of decline is similar to the other cores and so it crosses PI values (GI = 0) later than the Vostok but earlier than the EDC curves, between 404 ka and 403 ka.

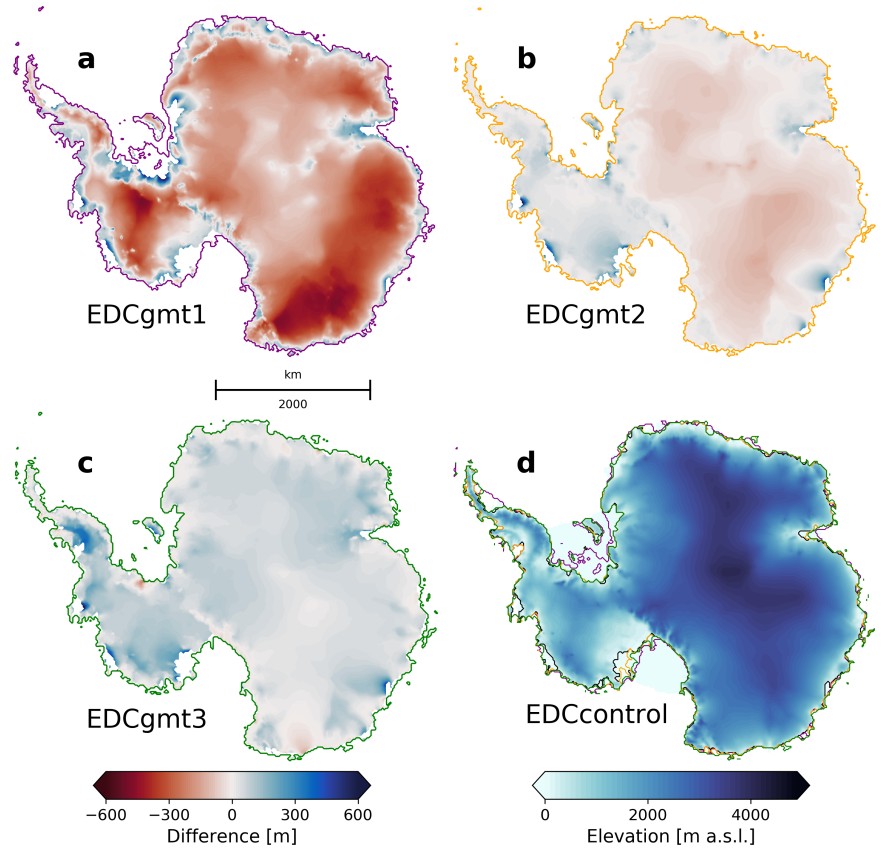

**Figure 3.** (a-c) Three different starting ice sheet geometries at 420 ka for gmt1–3 using EDC forcing. The EDC CFEN member is used as "control". The same spatial pattern is seen for DF and Vostok cases, and the averaged ice elevation difference between their respective geometries amounts to less than 50 m. Color scheme shows differences in surface elevation between each geometry and the control for 420 ka (d). Differences are only shown where the ice is grounded in both geometries, and coloured lines show the respective grounding lines in gmt1-3, also overlain in (d)

The ice sheet history for MIS11c using the LR04 forcing is clearly different from the others. The ice sheet loses less than a third of its volume compared to the other CFEN members, and becomes smaller than PD for a duration of 9 kyr, while the others are consistently below PD levels (Fig. 4a). It is worth reminding that, in contrast to other members of CFEN, the LR04 curve starts with colder-than-PI conditions and does not produce a peak warming as strong as the others. It only shows a brief period of warmer-than-PI conditions between 410 and 401 ka (Fig. 2b), resulting in an overall larger AIS (Fig. 5). The ice core CFEN members yield lower ice volumes throughout the entire MIS11c (Fig. 4a), but with important variations. The Vostok-forced experiment, for example, suffers a faster ice loss at the beginning of the simulation period, when it shows a

**Table 3.** Summary of performed experiments grouped by ensemble, listing their respective GI forcings, applied sea level reconstruction, and choice of initial geometry. LGMavg denotes that the GI was rescaled using the average LGM value as opposed to the peak value (cf. Sect. 2.3.1 and Table 4). The SGSEN experiments were grouped for better visualisation, but each SGSEN row corresponds to 3 experiments, one starting from each geometry (gmt1–3).

| Ensemble | Experiment | GI forcing | Sea level reconstruction | Initial Geometry |
|---|---|---|---|---|
| CFEN | lr04 | LR04 | Bintanja and van de Wal (2008) | control |
| CFEN | edc | EDC | Bintanja and van de Wal (2008) | control |
| CFEN | df | DF | Bintanja and van de Wal (2008) | control |
| CFEN | vos | Vostok | Bintanja and van de Wal (2008) | control |
| SSEN | lr04lgmavg | LR04$_{\text{LGMavg}}$ | Bintanja and van de Wal (2008) | control |
| SSEN | edclgmavg | EDC$_{\text{LGMavg}}$ | Bintanja and van de Wal (2008) | control |
| RSEN | lp1bx | EDC (1 kyr low pass, LP) | Bintanja and van de Wal (2008) | control |
| RSEN | lp3bx | EDC (3 kyr low pass, LP) | Bintanja and van de Wal (2008) | control |
| RSEN | lp5bx | EDC (5 kyr low pass, LP) | Bintanja and van de Wal (2008) | control |
| SLSEN | s16l | EDC | Spratt and Lisiecki (2016) long | control |
| SLSEN | s16s | EDC | Spratt and Lisiecki (2016) short | control |
| SLSEN | s16u | EDC | Spratt and Lisiecki (2016) upper uncertainty | control |
| SLSEN | spm | EDC | Imbrie et al. (1989) | control |
| SLSEN | wae | EDC | Waelbroeck et al. (2002) | control |
| SGSEN | edcgmt[1-3] | EDC | Bintanja and van de Wal (2008) | gmt1-3 |
| SGSEN | dfgmt[1-3] | DF | Bintanja and van de Wal (2008) | gmt1-3 |
| SGSEN | vosgmt[1-3] | Vostok | Bintanja and van de Wal (2008) | gmt1-3 |

sudden warming. However, it recovers more quickly than the EDC and DF experiments as soon as the peak warming is over and the climate starts to shift back to PI conditions, without a WAIS collapse (we consider the WAIS to have collapsed when the Weddell, Ross, and Amundsen seas become interconnected; Fig. 5).

The members that result in a collapse of the WAIS (forced with the DF and EDC reconstructions) reveal slightly different responses (Fig. 4a). The experiment forced by the EDC reconstruction shows an AIS volume reduction after a sudden warming at around 418 ka, but the WAIS collapse is delayed until 407–406 ka (Fig. 5), following a second short period with an increased warming rate after 412 ka, that leads up to the peak-warming of MIS11c. The DF experiment on the other hand is rather stable until 412 ka, when the climate starts warming towards its peak. Most of the retreat is triggered after the sudden temperature rise at 412 ka, as opposed to when the peak warming occurs.

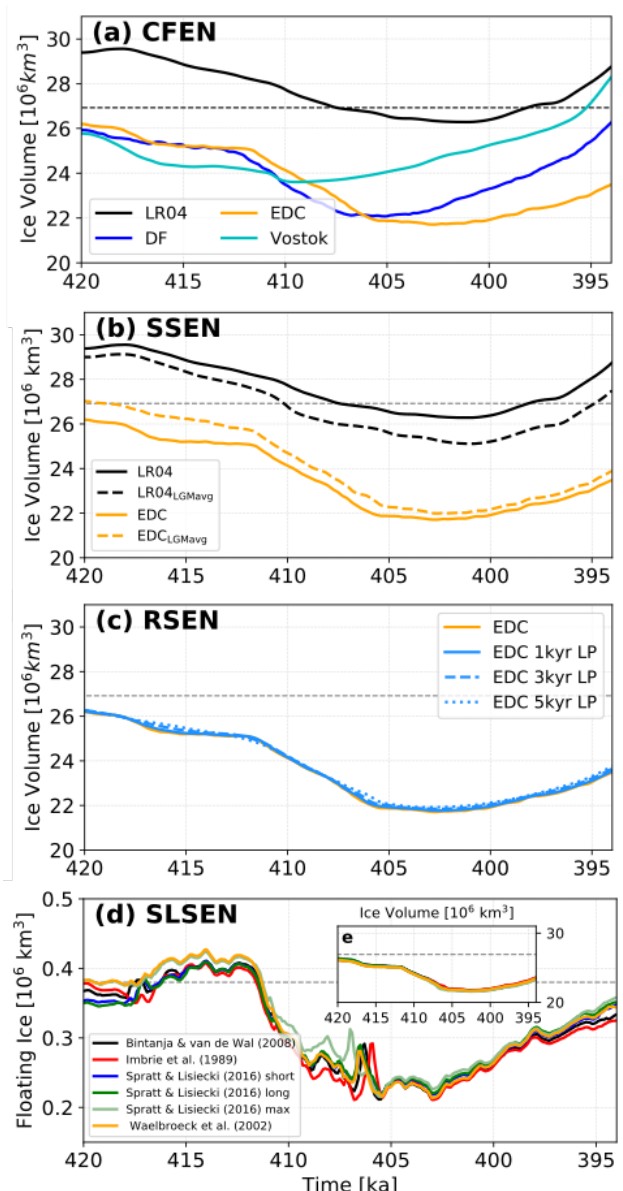

**Figure 4.** Sensitivity of AIS response (in total ice volume, $10^6\text{km}^3$) between 420 ka and 394 ka to (a) CFEN GI reconstructions; (b) SSEN rescaled GI reconstructions; (c) RSEN low-pass filtered GI reconstructions. Panels d and e show floating and total ice volumes (in $10^6\text{km}^3$), respectively, for the SLEN sea-level forcing reconstructions forced by EDC GI (cf. Table 3). Dashed line shows PD ice volume (Fretwell et al., 2013)

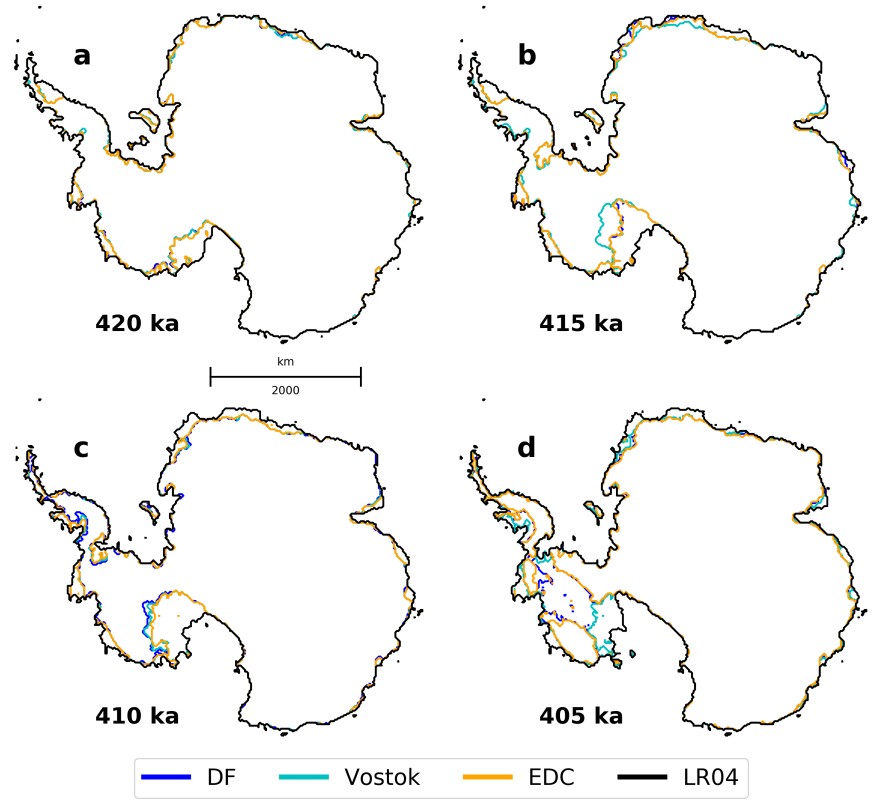

**Figure 5.** Grounding lines at 420, 415, 410, and 405 ka for the CFEN simulations.

## 3.2 Sensitivity to rescaling of the climate forcings

The different $\delta$ isotope reference values used for the SSEN experiments are shown in Table 4 (cf. Table 2). Using an LGM-averaged value results in a smaller ice sheet for the LR04 GI, while for the EDC GI it results in a slightly larger AIS than their correspondent CFEN experiments throughout the entire MIS11c (Fig. 4b). The LR04-LGM-averaged run, however, still does not produce AIS retreat as significant as the other experiments, with 4.2% less volume ($1.1 \cdot 10^6$ km$^3$) at 402 ka when compared to its original rescaling. The warmer conditions resulting from the GI rescaling are still not enough to compensate for the initial growth caused by significantly colder-than-PI conditions at 420 ka, and during the preceding relaxation stage. Although differences in ice-sheet volumes exist between the different scaling strategies in the EDC-forced experiments, the resulting ice sheet histories are quite similar. Despite ice-sheet volume at 402 ka being smaller in the run where the LGM reference is taken as the peak value, the differently scaled ice sheet is only 1.2% larger in volume than the CFEN ice-sheet ($0.3 \cdot 10^6$ km$^3$).

**Table 4.** Different isotope values adopted for the GI rescaling procedure. *LGMavg* is the reference value obtained from the average between 26 and 19.5 ka (which replaces LGM in Eq. 3 for the respective experiments; see Sect. 2.3.1).

| Record | $\delta X_{PI}$ [‰] | $\delta X_{LGM}$ [‰] | $\delta X_{LGMavg}$ [‰] |
|--------|---------------------|----------------------|--------------------------|
| EDC    | -397.4              | -449.3               | -442.3                   |
| LR04   | 3.23                | 4.99                 | 4.85                     |

## 3.3 Sensitivity to millennial variability and sea level reconstructions

The trajectories of each ensemble member in RSEN agree with one another (Fig. 4c), showing increased delays in the ice sheet retreat in response to the filtering intensity. Also, although it is possible to see slight differences in ice sheet volumes between ensemble members (the volume is larger the more filtered the forcing is), it is negligible compared to the overall changes in volume experienced by the entire ensemble.

Although the range of global mean sea level reconstructions is wide (nearly reaching 60 m between 405 ka and 400 ka; Fig. 2c), the AIS response in terms of volume is remarkably similar for different sea level curves (Fig. 4e). The differences in sea level have their largest impacts on the volume of floating ice (Fig. 4d). Thus, floating ice volume directly reflects the sea level forcing effect on the flotation of ice, and consequently on the grounding line position. The SLSEN member with the highest sea level rise (i.e., the upper uncertainty boundary of Spratt and Lisiecki, 2016) deviates the most from the other members, especially in the portion of grounded ice being brought to flotation (Fig. 4d). However, the differences are not significant enough to yield substantially distinct ice volume changes (Fig. 4e).

## 3.4 Sensitivity to the choice of initial ice sheet geometry

Looking at how the four initial geometries (gmt1-3 and the control) evolve under the three different climate forcings from the ice-core derived GI reconstructions (Fig. 6), it becomes clear that all members under the same climate forcing have a tendency to follow the same path despite differing initial ice sheet configurations. The spread in minimum ice-sheet volumes (and consequently implications for WAIS collapse) due to assumptions of starting geometry becomes rather small, between 1 and 3 m s.l.e. at 405 ka among the three different forcings in SGSEN. The different ice sheet configurations also show a similar pacing of retreat after 412 ka, indicating that their corresponding volume by that time did not affect its rate of retreat due to climate warming. In our SGSEN simulations, it appears that the main source of variability between ice sheets with different initial geometries comes from specific EAIS drainage basins, such as those of Cook, Totten, and Dibble glaciers (Fig. 7 showcases the EDC ensemble; cf. Fig. 1 for geographical locations). The latter two remain thicker in the alternative geometry experiments than in the correspondent CFEN experiment, whereas the former is thinner in gmt3 (Fig. 7c). Some variability can also be observed in the WAIS domain. Parts of Pine Island Glacier appear to resist ice sheet collapse in the thicker-ice-geometry experiments (gmt3) when compared to the CFEN-equivalent run (Figs. 7c,d). Given the observed spread, the three ensemble members constrain the range of potential sea level contributions from Antarctica during the MIS11c highstand to 4.0–8.2 m

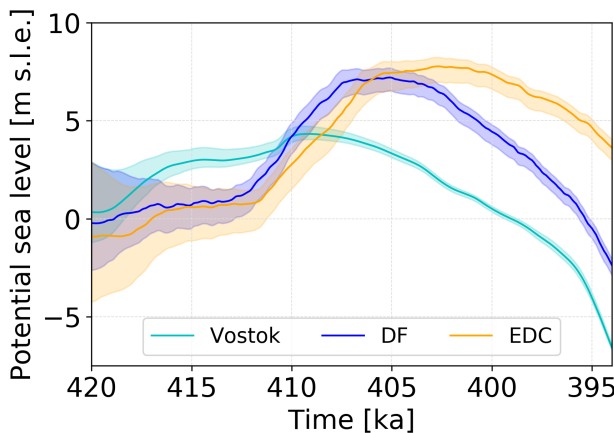

**Figure 6.** Sensitivity of the AIS response to CFEN GI reconstructions (Vostok, DF, EDC) between 420 and 394 ka with uncertainty bands from four distinct initial ice sheet starting geometries (gmt1–3 and respective CFEN member), expressed in contribution to global mean sea level [m s.l.e.]. Solid lines show the mean of each common-forcing ensemble member, while the color filling shows the spread given by the different starting geometries.

(minimum from Vostok at 410 ka, maximum from EDC at 405 ka). This range of 4.2 m essentially corresponds to whether the WAIS has collapsed or not during MIS11c.

## 4 Discussion

Our simulations show that during the peak of MIS11c, the WAIS probably collapsed. We base this statement on results from
experiments forced by different proxy records with significant differences in their structure during the MIS11c peak warming. One consisted of a short single peak (Vostok), while others showed a prolonged period of (relatively) warmer conditions (LR04, DF, and EDC). Despite having a warming peak of a similar GI magnitude at 410 ka, the Vostok-forced CFEN member is the only ice core-forced ensemble member that shows no collapse of the WAIS. Although the remaining climate reconstructions all show a longer peak, differences still exist among them. For example, EDC and DF, which are the most similar to each
other, start shifting to their warmest conditions at about the same time around 414 ka, but peak at different times. DF peaks at 410 ka, which is 3 kyr earlier than EDC. Regardless of this difference, the simulated WAIS collapse occurs at 407 ka using the DF and at 406 ka using the EDC core forcing, which is closer than their timing of peak warming. Experiments forced by both records also yielded similar ice volumes (Fig. 4a) and extents (Fig. 5). It should be mentioned that the combination of GI and climate-model forcing results in a warmer signal in the surface temperatures at the DF, EDC, and Vostok core sites than
obtained directly from their $\delta D$ records (Supplementary Fig. S14). This is most likely due to the LGM cold bias in CCSM3, which persisted despite the lapse-rate correction applied. Since PI temperatures do not have any strong bias, the LGM cold bias causes the GI reconstruction to yield colder temperatures during colder-than-PI times (GI > 0), and warmer temperatures

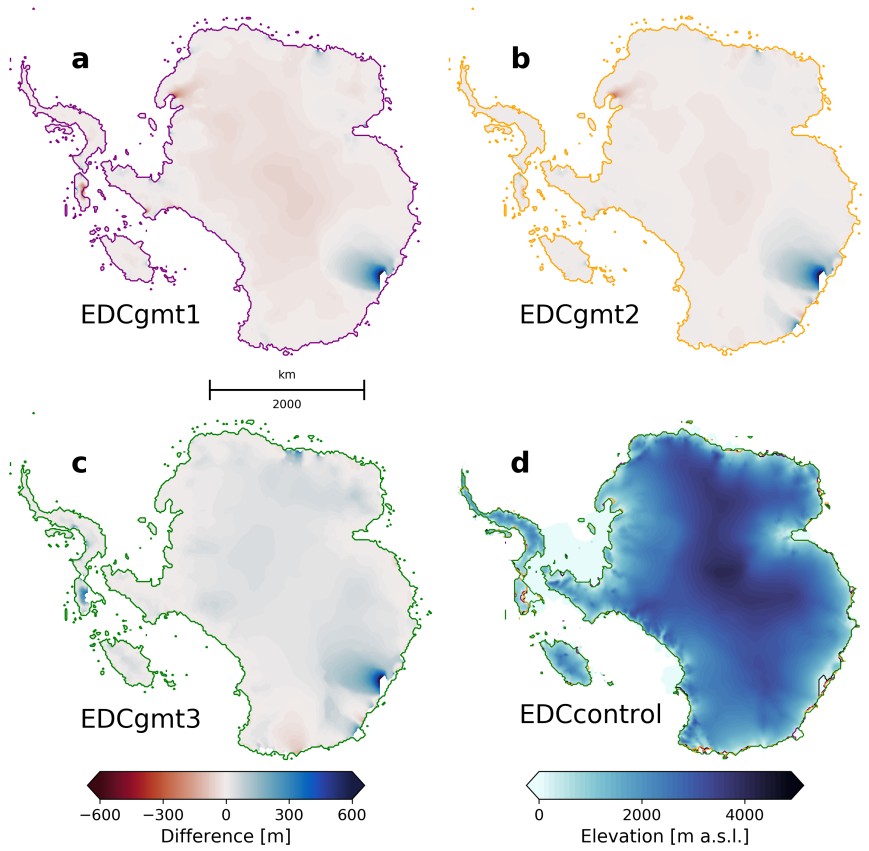

**Figure 7.** (a-c) Ice sheet geometries at 405 ka for the EDC CFEN member using three different starting geometries at 420 ka (Fig. 3). Color scheme shows differences in surface elevation between each geometry and the control for 405 ka (d). Differences are only shown where the ice is grounded in both geometries, and coloured lines show the respective grounding lines in gmt1-3, also overlain in (d)

during warmer-than-PI times (GI < 0). Nevertheless, Vostok's GI-reconstructed temperature peak matches the peak observed in DF for its $\delta$D-derived curve, and is also close to the warmest temperature reconstructed with the EDC isotopes. Finally, LR04 315  stands out when compared to the ice cores, and will be discussed in more detail separately.

Although sensitivity experiments show WAIS-collapse results using DF and EDC to be robust, the timing of the events discussed above should be taken with caution for two main reasons. First, we are forcing the entire AIS model with a climate signal from the EAIS, while previous studies have shown that the WAIS could have responded over 2 kyr earlier to changes in climate (WAIS Divide Project Members, 2013). Second, all discrepancies in the timing of the events discussed so far recorded 320  by the ice-core records, especially the peak warming and ice sheet collapse, are within the uncertainty in their respective age

models (Parrenin et al., 2007; Bazin et al., 2013). Consequently, these two factors prevent us from establishing an exact timing of these events, which means that the lags in AIS response are the most important to be considered.

In all our CFEN simulations, ice sheet retreat is associated with stronger basal melting close to grounding lines, especially at Siple Coast, and in the Ross and Filchner-Ronne ice shelves (Fig. 8). Surface ablation seems to be significant only over the fringes of the EAIS, notably at Dronning Maud Land (DML) and the Amery ice shelf, where surface temperatures reach positive values during summer (Fig. 9a). Nevertheless, they show limited retreat compared to the aforementioned WAIS ice shelves. The strong WAIS retreat seen in the EDC and DF-forced runs starting from 412 ka is triggered by an increase in ocean temperatures at intermediate depths (hereafter defined as the average between 400 and 1000 m depth) under the Ross and Filchner-Ronne ice shelves (Fig. 9b). Although this increase is progressive, it triggers a faster loss of volume by the WAIS compared to the EAIS after 412 ka (Fig. 9c), in contrast with a similar evolution between the ice sheets before then. This observed tipping point at 412 ka also explains why the different initial ice-sheet configurations under a common forcing follow the same trend from that moment onwards (Fig. 6), and why the evolution of WAIS and EAIS sea level contributions diverge. As ocean forcing becomes the main driver of ice-sheet retreat, it has a much larger impact on marine-based portions of the ice sheet. Around most of the EAIS (except for the Amery Ice Shelf), ice shelves are small and provide little buttressing. Hence, because most of the EAIS is grounded above sea level, its sub-shelf melting is not high enough to force grounding line retreat as strongly as in the WAIS. As a consequence, ice melt is dominated by surface ablation at the ice-sheet fringes (cf. hatched patterns in Fig. 8).

The average intermediate-depth ocean temperatures under the Filcher-Ronne and Ross ice shelves peak between 0.4 and 0.85 °C for the three ice core-forced CFEN members (Fig. 9b). This happens at 410 ka for Vostok, 408 ka for DF, and 407 ka for EDC. Strong WAIS retreat, however, starts before the peak in forcing, supporting the presence of a tipping point at 412 ka. To further test whether this tipping point is the trigger of WAIS collapse, we have performed four additional experiments: *(i)* forced by EDC GI, but keeping the GI constant after 416 ka (i.e., before the threshold found in ocean temperatures), *(ii)* forced by EDC GI, but keeping the GI constant after 410 ka (i.e., just after the sudden increase in ocean temperatures, but before the maximum is reached; cf. Fig. 9b), *(iii)* forced by Vostok GI, where climate forcing is kept constant at its peak condition at 410 ka, and *(iv)* forced by Vostok GI where, after the 410 ka peak, GI is brought back to its 411 ka value (i.e., between the peak and the observed tipping point) and kept constant. Figures 10a,b show that keeping the EDC-derived climate constant at 416 ka conditions prevents the WAIS from collapsing, while keeping it constant at 410 ka conditions delays its collapse by almost 5 kyr compared to the core CFEN run. The Vostok-based simulations (Figs. 10e-h) show that there is indeed a threshold in ocean temperatures, which is of approximately 0.45 °C for the Filchner-Ronne ice shelf, and 0.54 °C for the Ross ice shelf. However, our results also imply that this threshold must be sustained for at least 4 kyr to cause a collapse (compare red and blue dashed lines in Figs. 10f-h). A short peak at this threshold and subsequent cooling prevents the WAIS from collapsing, compared to keeping it constant at the same peak value (Fig. 10e,f). Comparing these values to PI temperatures averaged over the same extent of the water column, the magnitude of warming necessary to cross this threshold is 0.4 °C. In other words, a warming of this magnitude can be understood as the condition necessary for WAIS collapse (Figs. 10c,d,g,h). Additional experiments where we test for a weakened ocean forcing further confirm this threshold, as a complete collapse of the WAIS is prevented

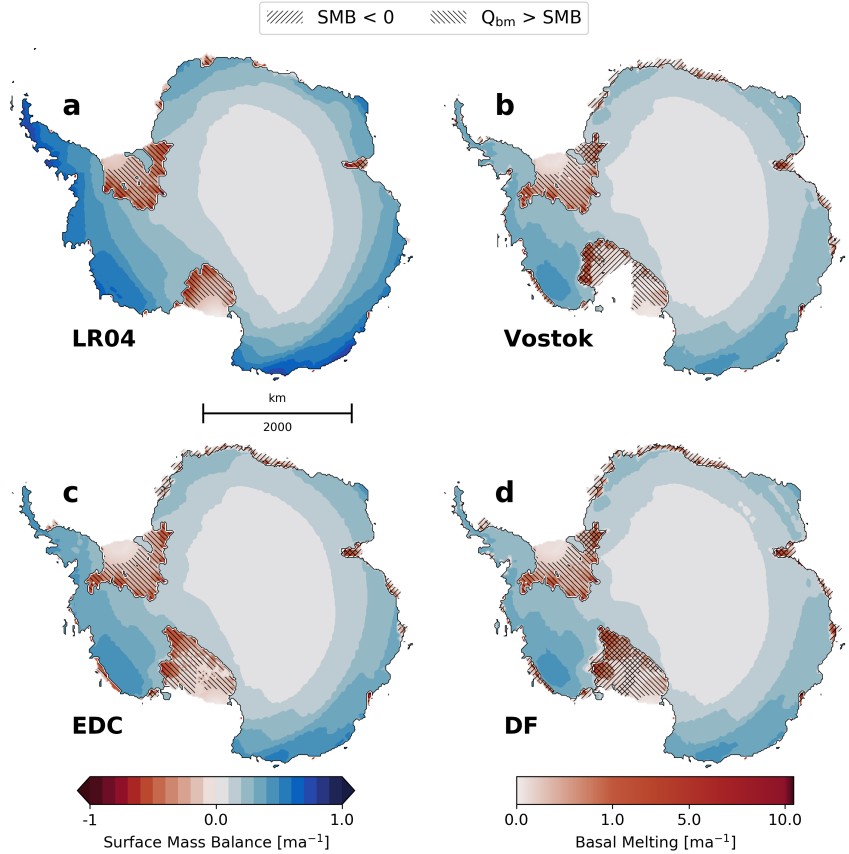

**Figure 8.** Surface Mass Balance (SMB, ma$^{-1}$) for the grounded ice and basal melting (Q$_{bm}$, ma$^{-1}$) for the ice shelves for the CFEN simulations at 415 ka. Hatched areas show where basal melting dominates over surface mass balance and where surface mass balance is negative (i.e., where surface ablation occurs).

when the temperatures at intermediate depths fail to reach a 0.4 °C warming relative to PI under the Filchner-Ronne and Ross ice shelves (Sect. 4 of the supplementary material). Considering that the temperature peak reconstructed by the Vostok GI is the closest to the $\delta$D-derived temperature peaks in DF and EDC (Fig. S14), a more prolonged warming as seen in the DF and EDC ice core seems to be a crucial condition for the modelled WAIS drawdown during MIS11c. For example, if the GI-derived

temperature for DF was not overestimated, and had its peak value close to its isotope-derived value, the response would likely resemble the experiment where Vostok-peak conditions were kept constant from 410 ka onwards.

The inferred critical warming of intermediate-depth ocean temperatures of 0.4 °C for MIS11c is close to the equilibrium model results in Garbe et al. (2020), but lower than results from Turney et al. (2020) for the AIS retreat during the LIG. While the former study shows a strong WAIS retreat is already possible for an ocean warming of 0.7 °C, the latter identifies a tipping

point at 2 °C warming in ocean temperatures. In other interglacials, such as the LIG, the shorter duration but higher intensity

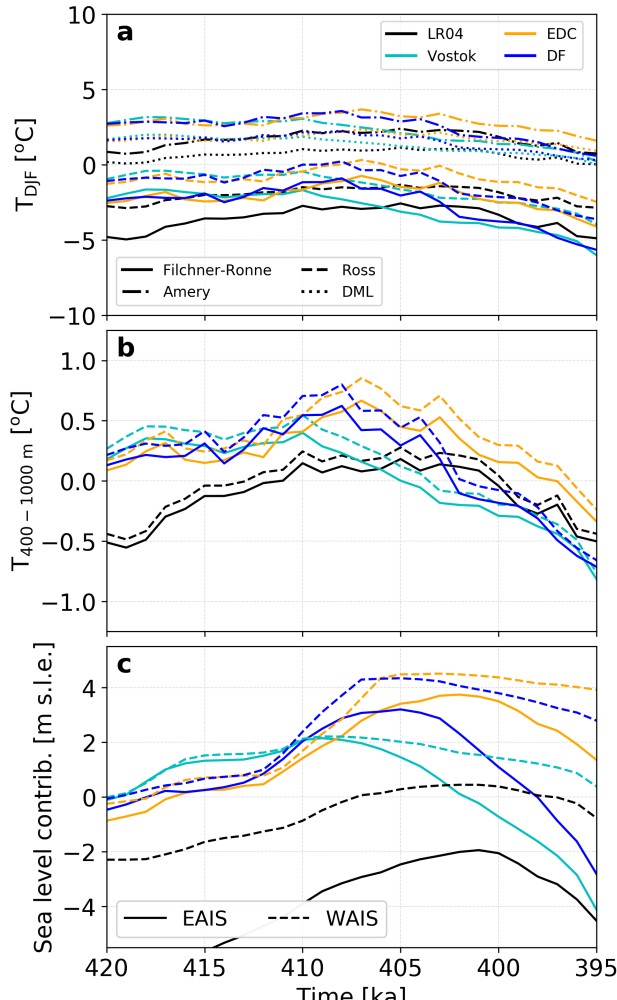

**Figure 9.** Evolution throughout MIS11c for each CFEN member for (a) Summer surface air temperature [°C] averaged over the main Antarctic ice shelves; (b) ocean temperatures averaged between 400 and 1000 m [°C] for the Filchner-Ronne and Ross ice shelves; (c) sea level contribution by EAIS and WAIS. Colours denote the respective CFEN member, while line styles in panels (a,b) denote each ice shelf, and each ice sheet in panel (c). DML refers to all smaller ice shelves along the Dronning Maud Land margin.

of ocean warming compared to MIS11c could have triggered WAIS collapse (Dutton et al., 2015; Turney et al., 2020), since a stronger rate of warming can drive ice retreat at a much faster pace. Thus, WAIS collapse during MIS11c was likely attained because ocean temperatures exceeded a modest threshold for long enough (over 4 kyr).

Despite differences in the model sensitivity to ocean temperature, our results support those of Tigchelaar et al. (2019)
and Albrecht et al. (2020) regarding the minor role that variations in sea level play in driving ice-sheet retreat compared to other external forcings. Although the coarse treatment of the grounding lines could have had an influence on the seeming

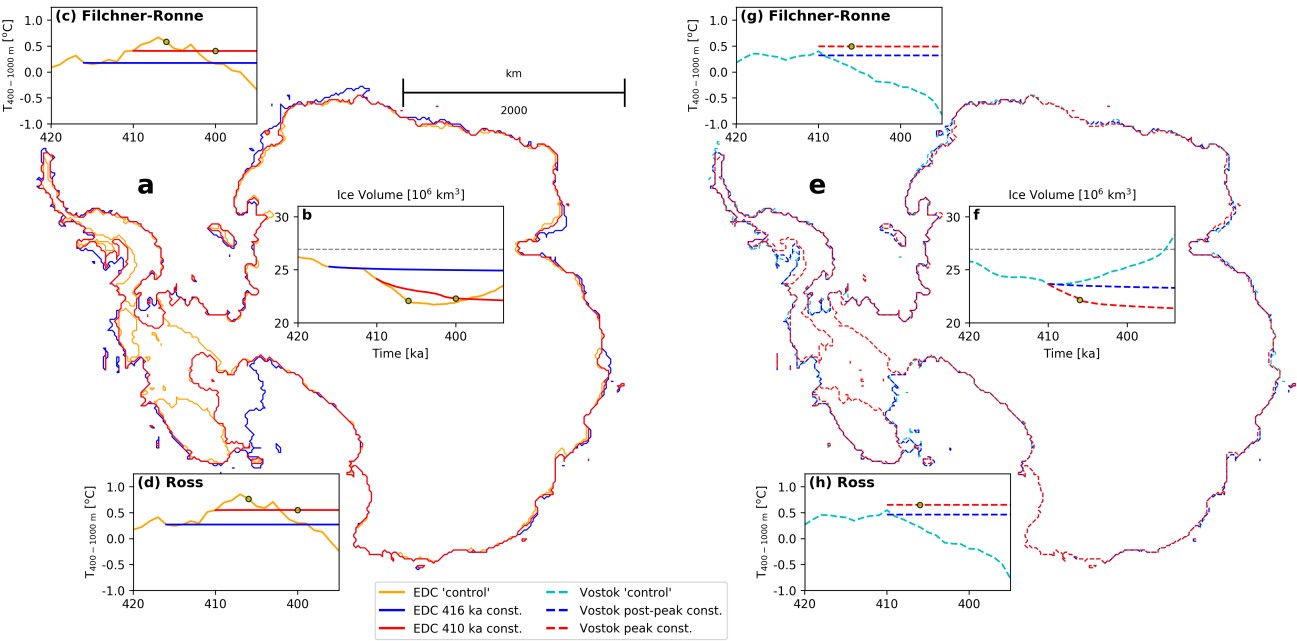

**Figure 10.** Thresholds for WAIS collapse. (a,e) grounding lines at 405 ka for three EDC-based (solid lines) and three Vostok-based (dashed lines) experiments, respectively (see below for explanation); (b,f) ice volume ($10^6$km$^3$), (c,d; g,h) intermediate-depth (400–1000 m) ocean temperatures [$^\circ$C] for the Filchner-Ronne and Ross ice shelves, respectively. Time series cover the period between 420 and 395 ka for both EDC (solid lines) and Vostok-based (dashed lines) experiments. Orange line shows the EDC control run, while cyan line shows the Vostok control run. Blue lines show EDC and Vostok simulations where climate was kept constant and the WAIS did not collapse, while the red lines show EDC and Vostok simulations where climate was kept constant and the WAIS collapsed. Yellow circles show the moment when the WAIS breaks down and an open-water connection between the Ross, Weddell and Amundsen seas is established.

insensitivity of our experiments to sea-level uncertainties, other models of similar resolution which apply different sub-grid parameterisations to the grounding lines yield similar results (Tigchelaar et al., 2019; Sutter et al., 2019; Albrecht et al., 2020). Hence, while this caveat must be taken into consideration, it does not appear to have influenced our results dramatically.

Moreover, AIS minimum extent and the timing of WAIS collapse are robust regardless of model resolution (Fig. S15). A set of simulations performed with several resolutions (from 20 to 10 km) showed virtually the same changes in ice-sheet extent, and modest variations in ice volume, which amount to a spread of 1.2 m s.l.e. in sea level contribution at 405 ka. Alternative sliding laws or sub-shelf melting parameterisations, for example using a linear dependence of sub-shelf melt to ocean thermal forcing, or applying a more physically realistic approach (e.g., Reese et al., 2018) were not tested, and could influence our

results. For example, numerical modelling studies in which the WAIS did not collapse during MIS11c were acknowledged to be less sensitive to the ability of ocean temperatures to drive basal melting (Pollard and DeConto, 2009; Tigchelaar et al.,

2019). Finally, we note that, despite very different approaches in reconstructing transient signals, neither Pollard and DeConto (2009) nor we were able to simulate a collapse of the WAIS using the LR04 stack as climate forcing.

The LR04 reconstruction is composed of a stack of 57 globally-distributed ocean sediment cores (Lisiecki and Raymo, 385 2005), with a strong deficit over the Southern Ocean. In the Nordic Seas, paleoceanographic records indicate that the ocean was colder than present during MIS11 (Bauch et al., 2000; Kandiano et al., 2016; Doherty and Thibodeau, 2018). Colder ocean temperatures in the Northern Hemisphere explain why LR04 shows oxygen isotopic values similar to the Holocene during MIS11c (Lisiecki and Raymo, 2005) despite the geological evidence that there was a contribution to higher-than-Holocene sea levels from both Greenland and Antarctica (Scherer et al., 1998; Raymo and Mitrovica, 2012). Hence, the inclusion of many 390 Northern Hemisphere records in the LR04 stack explains why it fails to capture the Antarctic warming during MIS11c seen in the ice cores, and the differences in timing compared to them. This also helps explain why the different criteria adopted for changing its scaling procedure had little effect on the results (Fig. 4b). A possible way of circumventing this problem could be to adopt a similar scaling approach to Sutter et al. (2019), who combined the LR04 stack and EDC ice-core temperature records, which, in their study, also led to WAIS collapse during MIS11c.

395 In East Antarctica, our simulations do not capture the ice sheet retreat into the Wilkes Subglacial Basin recently proposed by Wilson et al. (2018) and Blackburn et al. (2020) for MIS11. Blackburn et al. (2020) suggest this retreat to have been caused by ocean warming, with little to no atmospheric influence. However, further paleoceanographic data are needed to fully understand this retreat (Noble et al., 2020), which so far has not been captured by other model experiments (cf. Wilson et al., 2018, Fig. 2b). As for West Antarctica, far-field sea level reconstructions suggest that a WAIS collapse was the most probable scenario (Raymo 400 and Mitrovica, 2012; Chen et al., 2014) when comparing global highstand estimates with the probable contribution from the GIS. While Robinson et al. (2017) found that Greenland contributed between 3.9 and 7.0 m to sea level rise (having 6.1 m s.l.e. as the most likely value), the AIS contribution cannot be constrained by simply subtracting the GIS's contribution from the global sea level highstand. The suggested asynchronicity between the GIS and AIS minimum extents (Steig and Alley, 2002) and the uncertainties in the age models of the different analysed ice cores (Petit et al., 1999; Parrenin et al., 2007; Bazin et al., 405 2013) prevent a simple relationship between both ice-sheet records to be established. Based on the ice-core experiments, our range for the potential sea level contribution of the AIS is 4.0–8.2 m. This wide range is mainly related to whether the WAIS collapses or not. Considering the cases where the WAIS collapsed (i.e., EDC and DF ice core experiments) as the most probable scenario, our range for the potential sea level contribution of the AIS is 6.7–8.2 m. In this case, the EAIS contribution is the largest source of uncertainty, being most sensitive to the choice of starting ice geometry. This effect is strongest over Wilkes 410 Land, where the spread in position of the grounding line is wider, and ice thickness is more variable than for other basins (Fig. 7). While nearby drainage basins, such as those of Totten and Dibble glaciers, become more stable given the larger ice sheet configurations of the alternative geometries (Figs. 3b,c), Cook glacier, emanating from Wilkes Subglacial basin, appears to thin regardless of the choice of initial geometry (Figs. 7a-c). Overall, the EAIS contributes 1.7 to 3.7 m s.l.e. during the highstand (Fig. 11). Conversely, the WAIS was rather insensitive to the choice of starting geometry (yielding 4.3–4.5 m s.l.e. 415 during the highstand in the case of a collapse, and 2.0–2.2 otherwise) due to the stronger role played by the sub-shelf ocean forcing after 412 ka. There are, however, two stabilising feedbacks which are not incorporated in our model: *(i)* a local sea-level

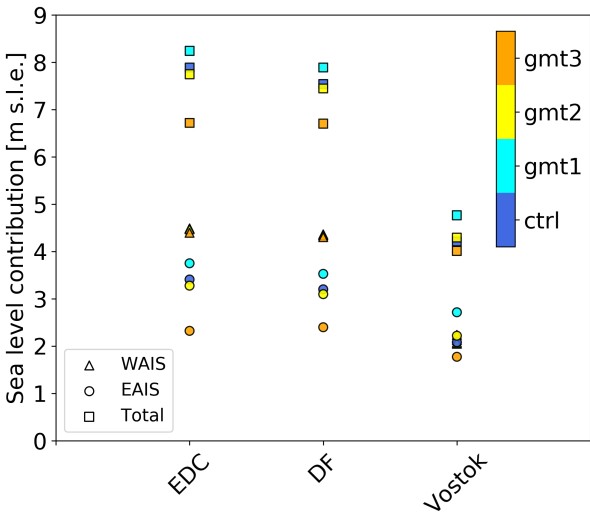

**Figure 11.** Sea level contribution (in m s.l.e.) of each SGSEN member during the global sea level highstand (405 ka for EDC and DF, 410 ka for Vostok).

drop caused by a reduced gravitational attraction of a shrinking ice sheet (e.g., Mitrovica et al., 2009), and *(ii)* the observed faster rebound of the crust due to a lower mantle viscosity in some WAIS locations (Barletta et al., 2018). The first effect is probably small based on our model's insensitivity to sea-level changes over these time scales, but we have been unable to

robustly test the effect of a faster rebound on AIS response during MIS11c. However, we note that our ELRA model is set up with a relatively short response time of 1 kyr, for which the resulting bedrock uplift is still not able to trigger a stabilizing effect large enough to prevent WAIS collapse.

## 5   Conclusions

Several studies have been carried out in order to reconstruct past ice changes over the Antarctic continent, but to our knowledge

no special focus has been given to Antarctica's response to the peak warming during MIS11c and the driving mechanisms behind it. To fill this gap we evaluated the deglaciation of Antarctica using a numerical ice-sheet model forced by a combination of climate model time-slice-forcing and various transient records through a Glacial Index (GI). The records were obtained from ice cores of the EAIS interior and a stacked record of deep-sea sediment cores taken from far-field regions. We evaluated the sensitivity of our results to *(i)* the scaling of the GI, *(ii)* millennial variability and temporal record resolution, *(iii)* different sea

level reconstructions, and *(iv)* initial ice sheet configurations. While sea level, higher-frequency variability, and the GI scaling of the records seemed to play a small role, different responses were seen for both East and West Antarctic Ice Sheets regarding the different applied transient signals, and for the initial ice sheet configurations. Among the applied ice-core reconstructions, the warming captured by the Vostok ice core during MIS11c was not strong enough to cause a collapse of the WAIS, which was

attributed to the short duration of its peak. Our results indicate that our modelled WAIS collapse was caused by the duration rather than the intensity of warming, and that it was insensitive to the choice of the starting geometry. The latter proved to be a larger source of uncertainty for the EAIS. Regarding the initial questions posed in the beginning of this study, we now provide short answers to them:

1. **How did the AIS respond to the peak warming of MIS11c? What are the uncertainties in the AIS minimum configuration, its timing and potential sea level contribution?**

   Using transient signals from EAIS ice cores, we found a range in sea level contribution of 4.0 to 8.2 m s.l.e., which mainly reflects whether the WAIS has collapsed or not in our experiments. For the former scenario –which is supported by far-field sea level reconstructions– we find that a WAIS collapse during MIS11c is attained after a prolonged warming period of the ocean of ca. 4 kyr. The resulting AIS contribution in this case is 6.7–8.2 m s.l.e. at 405–402 ka. Uncertainties in these values are primarily due to the choice of climate forcing and ice sheet starting configuration (at 420 ka). While the contribution to sea level rise by the WAIS was consistent among those experiments that yielded its collapse (4.3–4.5 m s.l.e.), the EAIS contribution remained more uncertain because of its sensitivity to the initial geometry of the ice sheet (2.3–3.7 m s.l.e.).

2. **What was the main driver of the changes in the AIS volume? Was it warming duration, peak temperature, changes in precipitation, or changes in the oceanic forcing?**

   We identify a tipping point at ca. 412 ka, beyond which strong WAIS retreat occured in response to the ocean warming. Past this point, retreat leading to WAIS collapse was mostly sensitive to warming duration more than intensity, provided ocean temperatures at intermediate depths become 0.4 °C warmer than PI under the Filchner-Ronne and Ross ice shelves. We found that this threshold needed to be sustained for at least 4 kyr for strong WAIS ice retreat to be triggered.

*Code and data availability.* The numerical code for the ice-sheet model SICOPOLIS can be obtained in http://sicopolis.net/. All settings files used for the model runs are available in https://github.com/martimmas/MIS11c_exps. The full model outputs are available upon request to the corresponding author.

*Author contributions.* MMB, IR and JB designed the study. Experiments were carried out and analyzed by MMB and JB. MMB wrote the manuscript with contributions from all co-authors

*Competing interests.* The authors declare that they have no conflict of interest.

*Acknowledgements.*  This work is funded by the MAGIC-DML project. MAGIC-DML is a consortium supported by Stockholm University (Arjen Stroeven), Norwegian Polar Institute/NARE under Grant "MAGIC-DML" (Ola Fredin), the US National Science Foundation under Grant No. PLR-1542930 (Jonathan Harbor & Nathaniel Lifton), Swedish Research Council under Grant No. 2016-04422 (Jonathan Harbor & Arjen Stroeven), and the German Research Foundation (DFG) Priority Programme 1158 "Antarctic Research" under Grant No. 365737614 (Irina Rogozhina & Matthias Prange). Jorge Bernales has been supported by the MAGIC-DML project through DFG SPP 1158 (RO 4262/1-

6). We would also like to acknowledge support from the Carl Mannerfelts fond and the Bolin Centre Climate Research School (Martim Mas e Braga). The ice-sheet model simulations were performed on the GeoMod cluster at MARUM, Bremen University. We thank Andreas Manschke for technical support and continuous access to the computer cluster.

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
