# Peer review of "Sensitivity of the Antarctic ice sheets to the warming of Marine Isotope Substage 11c"

_The Cryosphere, 2020_

## Referee Comment (RC1) · Clemens Schannwell (Referee) · 10 Jun 2020

**Review of Mas e Braga et al. "Sensitivity of the Antarctic ice sheets to the peak warming of Marine Isotope Stage 11"**

June 10, 2020

**General comments:**

The manuscript by Mas e Braga et al. presents simulations with the ice-sheet model SICOPOLIS of the Marine Isotope Stage 11c (MIS11c). This interglacial period is close to pre-industrial conditions in terms of orbital parameters and therefore a good analogue for present-day conditions. The authors perform a number of sensitivity experiments in which they investigate the influence of different parameters (e.g. sea-level, initial geometry etc.) on the evolution of the Antarctic ice sheet. They also attempt to identify key processes that may also be relevant for future ice-sheet changes.

I find the topic and the idea of the paper interesting and the results should be of interest to the cryospheric modelling and paleoclimate community. Overall, I think the manuscripts presents enough novelty and hence merits publication. However, the presentation of the results in its present form is not ready for publications and I recommend the authors take into account my comments listed below. I hope the authors find my comments helpful.

**Specific comments:**

**Main concerns:**

1. In its present form, I find the manuscript hard to follow in many places (more in the technical corrections below) and really difficult to judge the results of the simulations because there is almost no description of the ice-sheet model and its boundary datasets and conditions. I suggest to expand section 2.1 to add this required information. To be more specific, what type of stress balance does SICOPOLIS use? What kind of basal friction law do you apply? I know you list the parameters in Table 1, but without the corresponding equation, they are rather useless. Does your basal friction coefficient

vary spatially and/or temporally? What are your boundary conditions for your enthalpy equation (e.g. do you specifiy a geothermal heat flux? Is it spatially constant?)? How do you treat calving in the model? There are a number of ways how to parameterise this. Since you talk about this in your results, it is essential to know how this is handled in your model.

Also you should mention with what geometry your initialise your model. I believe it is with present-day geometry, but with which dataset (Bedmap2, Bedmachine)? Do you take the bedrock and ocean floor topography from the same dataset?

2. Could you please motivate the ensembles or parameters changes that you are investigating a bit more? As it stands now, it seems like you picked a number of parameters, but there also could an argument be made for a bunch of other parameters to be varied.

3. I find most of the figures (e.g. 4, 6,7,9) not very informative. Looking at integrated quantities is OK, but having five Figures like that is too much. I suggest to combine them into a Figure with several panels. I also find it hard to judge in these volume plots whether differences are small or large (Is 2000 km3 a lot?). Maybe better to plot it in percent normalised to your starting volume? Also just because your ice volume is similar does not mean you cannot have regional differences in grounding-line position or ice thickness. For example on P16L279 you state "... show similar retreat rates..." but I cannot find a Figure where this is actually shown. So I suggest to add some Figures, where we can also look at some spatial differences (a few suggestion in the technical corrections below). For example, you could plot some grounding-line positions from different simulations in 2D on top of each other to see the differences in retreat or lack thereof. I also encourage the authors to discuss their results more in depth. For example, they state in L276ff that different initial ice sheet configurations converge to the same geometry for the same climate forcings. This alone is quite surprising to me and at least warrants a discussion why potential feedback mechanisms (e.g. stabilising grounding-line on topographic height) are not triggered in these simulations?

4. I think you should scratch your attempt to identify drivers for future change. You have it in your research questions, but other than in the conclusion section you never mention it again. And your statement in the conclusion statement is extremely vague (and we know this already) and to be honest not backed up by your simulation results.

5. The abstract in its current form is much too long and too descriptive. Please shorten and make more concise.

6. This is more an optional point and maybe a matter of taste, but I think you could also add a model limitations section. There are a few places where you can shorten the main text (see below), so that this would not much increase the length of the manuscript.

I always find it helpful in modelling papers to have a section in which limitations and potential future avenues for improvements are discussed. I must admit that as the paper stands now with very little information about the ice-sheet model, it is hard to examine what the benefit of your model setup is?

**Technical corrections:**

Abstract:
L8 I do not think that the Greenland information is necessary in the abstract. Also the latter half of the sentence makes no sense to me "..., both configurations of the Antarctic ice sheets..."? What configurations?

L12 Does LR04 need to be introduced as an acronym? I did not know straight away what it is.

L17 Here and throughout, I find the term "ice-sheet contraction" unusual. I know what you mean, but I think more commonly it is referred to as "ice-sheet retreat". Please consider changing it.

L29-34 This sentence is way too long and confusing. Please split up and make clearer.

L43 What do you mean by "reduced stability"? And why would that trigger stronger glacial-interglacial cycles?

L49-55 I think this paragraph can be thrown out, as it is irrelevant to the Antarctic simulations in the paper. It suffices to say, I believe, that the ice-sheet history in Antarctica is more uncertain than for Greenland.

L56-58 The first half of the sentence is confusing. The way it is written, it makes it sound as if Raymo and Mitrovica estimated it to be 6-13 m above present-day? But why is there a reference to Dutton et al. then? Here and throughout, could you please try to keep sentences shorter. It makes it easier to follow for the reader.

L61-64 Again a very long sentence which I do not understand. Please break up the sentence and clarify.

L65-80 Here, I would like to see what your study adds to studies like the one from Sutter et al. 2019. What is the advantage of your study/model setup ?

L81 I do not agree that you are presenting model reconstructions. What you present are sensitivity experiments. But as far as I can tell, you are not trying to match any geological constraints which is what I understand as model reconstruction.

L85 As said above, I do not think you really address the last question about future ice-sheet changes. Therefore, I recommend removing it from the manuscript altogether.

L106 In addition to the changes suggested above. How do you initialise for the different ice-sheet configurations? Do you use the same temperature spin-up and let it evolve afterwards? Or do you let it evolve to a different geometry and do the temperature spin-up then with a fixed geometry?

L106 From where do you get your surface temperature distribution? An ice core only provides you with temperature changes with respect to a certain baseline. Please add this to this section.

L107-109 This means you just move this shock outside of your time period of interest? This is in general OK, but raises the following questions: What forcing do you apply for the 5 ka in which the ice geometry is allowed to freely evolve? And how far away do you get from your initial geometry? And I am also missing a plot where you show that your ice sheet is close to steady state. I would appreciate if you could add a plot for this.

L129, equation (2): From this equation I gather that you apply the same temperature differences to the ocean as you do to the atmosphere? And you also do not apply a time lag to the ocean warming/cooling? Is that really realistic giving the long response time of the ocean compared to the atmosphere? At the very least, this choice should be discussed somewhere in the text.

L137 To me all headers in this section should rather read "Model sensitivity to XXX". Because this is ultimately what you do in this paper, rather than rigorously quantifying uncertainties.

L139-141 This sentence needs rewriting. I am not sure I understand what you are saying.

L154 should be "mean sea level"

L166 Here I believe you say that you also initialise with present-day conditions, but this needs to come much earlier and with more info as to what datasets you used for this.

L221 you state: "... ice sheet contraction is associated with strong basal melting close to the grounding lines ...". First of all this comes a bit out of the blue. Secondly, you show little evidence that this is actually the case. In Fig. 5 you show that basal melting is dominating, but if you have different SMB rates, the basal melt rate could be either 1.5m/yr or 6 m/yr. Please also avoid relative terms like "strong" without giving any numbers. Do you mean 5, 50, or 500 m/yr when you say "strong" melting. Related to

this, do you apply melting to partially grounded grid cells or only to fully floating? This makes a big difference how much your grounding line retreats for similar melt rates.

L222 should read "... Siple Coast, at the Ross Ice Shelf, and underneath ..."

L223-224 & L227 Since your basal melt rate is a quadratic function of your ocean temperature, stating that it is a combination of warming of the upper ocean layer and high melt rates is saying the same thing. Please reformulate.

L228 Two things here. First, since you have a separate results and discussion section, I was expecting only a description of the results. However, here and in other places (e.g. L245, L256-259) in your results section you are interpreting and discussing your results already. So either you have a combined results and discussion section or you move this material to your discussion section. Secondly, I cannot confirm your statement that ice loss is dominated by surface ablation on Amery in Fig.5. First of all, the panels are too small, so I am not sure if Amery is hatched or not? I do not really understand the purpose of Fig. 5, but to me Amery looks pretty red which means to me that there is a lot of ablation in this area. So why would it not retreat there and why is ablation so high in this region compared to basal melting?

L237 "..., the resulting ice sheet histories are quite similar." This is true for the integrated ice volume, but again I find this quite superficial and it could be different when we look at 2D fields.

L242 If it is problematic why did you include it?

L256-259 This is discussion for me (see comment above).

L268-L274 This paragraph should rather be part of your experimental design section. By now there are so many simulations that you performed that I think it is really necessary to add a table where you list all the simulations with important forcing parameters in a table. It is really hard to keep track of the simulations.

L276 To me that is really surprising. From my experience, the initial geometry is quite important with regard to what your results look like at the end of the simulation. You glance over this, but this needs a discussion. Why do you think this is the case?

L279 "... also show similar rates of retreat ...". Again this is nowhere shown. I mean in Fig. 10 it looks like they actually have exactly the same grounding-line position. Is that true?

L282 Could you please add these locations to the respective Figure for better orientation.

L289-301 This is a weak introduction to the discussion and repeats most of the material that you covered in the introduction. Consider removing it.

L321 "... it seems that ice-shelf calving plays a role just as big". This again comes totally out of the blue and at the moment there is no way to check this statement as it is simply not described how calving is handled in the model. I also do not quite follow the explanation for this. Could the authors please elaborate on this?

L393 delete objective

L406-407 Delete last sentence (see comment above).

Comment hyphenation: I noticed that throughout the manuscript your use of hyphenation is inconsistent. You write ice-shelf calving, but then grounding line advance without hyphen. I am not sure what the TC policy is, but please make sure that you are at least consistent throughout the manuscript.

**Figures:**

Fig. 1: Please add a scale bar. Glacial index plots and their labels could be bigger.

Fig. 2: Why do you show the time series until present-day? I think a zoom in into the period of interest would be better.

Fig. 3: It is really hard to see any differences in the upper panel (a-d) with the current colour scale. Also the grounding-line position should be made more prominent (thicker line or different color). In general there is too much white space and subplot labels (a-g) are too small. Please make each subplot bigger for better readability. Please also add a scale bar.

Fig. 4: In the lower plot it looks like your model run for LR04 is not really in steady state or is your initial perturbation that large compared to your spin-up forcing? As mentioned above, I do not find the current y-axis units very intuitive for the lower panel. Labels (a,b) are too small.

Fig. 5: I do not really understand the point of this Figure as I do not get any information about the magnitudes of basal melting or the SMB. This Figure also needs a scale bar.

Fig. 6: See Fig. 4

Fig. 7: Labels (a,b) are too small.

Fig. 8: Labels (a,b) are too small.

Fig. 9: Labels (a,b) are too small.

Fig. 10: It is really hard to see any differences in the upper panel (a-d) with the current colour scale. Also the grounding-line position should be made more prominent (thicker line or different color). In general there is too much white space and subplot labels (a-g) are too small. Please make each subplot bigger for better readability. Please also add a scale bar. Consider using a different colour range for plots e-g as they are mostly white now and show little information. It would also good to add place information that you mention in the text. Ninnis and Totten glaciers.

Sincerely, Clemens Schannwell

---

## Referee Comment (RC2) · Anonymous Referee #2 · 18 Jun 2020

Braga et al. manuscript focuses on the Antarctic Ice Sheet contribution to MIS11. To understand what are the main driver of a potential AIS retreat during the MIS11c interglacial peak, they propose to explore the impact of climate forcing, initial geometry and global mean sea level history on AIS dynamics. They used a stand-alone SIA-SSA ice sheet model forced by idealized climate forcing: Pre-industrial climate is varied using the anomaly between Pre-Industrial and Last Glacial Maximum, scaled back to older times using Antarctic ice core records. Conclusions based on the simulations is that WAIS retreat is driven by the duration of the warmth during MIS11.

Although the aim of the manuscript is highly interesting, I find that many aspects have not been treated with care and part of the discussion is not conclusive because of the

lack of deeper analysis of the simulations and their forcing. It is clear that a tipping point appears in the simulations, probably resulting from oceanic forcing. Thus because of this, it is impossible to answer to the questions posed and impossible to conclude that the WAIS retreat int he simulations results from the duration of the interglacial during MIS11c. Some additional simulations are necessary to answer the 2 question posed in this manuscript. Especially, simulations testing the impact of the ocean forcing on the AIS retreat, thus by selecting lower ocean forcing. As they are, most of the results here could apply also for MIS5.

Many methodological aspects are not well described: methods section is poorly written, many details are missing about the ice sheet models (calving, sliding, sub-glacial hydrology, grounding line, surface mass balance) and about the forcing themselves. The generation of forcing deserve more figures in the supplementary since they are key to the conclusions of this manuscript. In the supplementary, comparison is made between simulated pre-industrial climate and present-day ERA5 fields, which is clearly incoherent. The generation of ocean forcing is also poorly documented here with again a comparison done between Pre-Industrial parameterized ocean forcing and ocean forcing inferred from observations ( I guess here Rignot et al, but this is not mentioned).

Analysis is sometimes superficial: no really discussion of the impact of oceanic forcing in the simulations, no comparison between the various terms of the mass balance through time (it could be done for different sectors of the AIS). Thus based only on the figures and analysis in the manuscript, many conclusion remains highly speculative. I think the manuscript has some potential. But substantial work is needed in addition to the one presented here before publication. Therefore I recommend major revisions.

General Comments:

I find inconclusive the set of experiments to determine whether or not the duration of the interglacial is responsible for AIS retreat rather than a warm peak as for MIS5. This is because the index derived from ice core records mainly impact on the oceanic forcing

of the simulation which generates a tipping point. Once the tipping point is crossed, then the duration of the interglacial does not matter at all to explain the amplitude of ice sheet retreat in the simulations. In simulations using Vostok-GI, ice volume is lower but because the GI does not yield too warm temperature. Thus the retreat is moderate. And in the case of EDC and DF, the retreat is comparable although DF presents a much longer peak for the interglacial. Actually, in question 2, ocean forcing is not mentioned at all as a potential driver of the ice sheet retreat. Same for ocean forcing: I would like to see a Figure in the supplementary of the oceanic forcing derived with the GI for the main ice shelves. You should discuss the impact to force WAIS with EAIS ice core records on the amplitude and timing of this retreat. For example, comparing them with WAIS divide ice core record. In those simulations, all forcing co-vary: your surface climate forcing and your oceanic forcing are modulated with the same index. It is likely not the case as atmosphere cools or warms faster than ocean does. This is not accounted for here. You could perform some interesting tests that would provide a nice discussion about the interaction between ocean and ice sheet. A plot showing air and ocean temperature forcing versus WAIS ice volume evolution; same for EAIS for all simulations is really necessary to support or explain better some aspect of this manuscript and provide answers to questions 1 and 2. Methods: in general the Methods are poorly explained, as well as boundary conditions (geothermal heat fluxes, bed topography etc...). Physics of the model is poorly described but some aspects are discussed within the manuscript. I would like to see a comparison between present-day simulated climate forcing and ocean forcing and observation from ERA5, and not between simulated PI and present-day ERA5.

Comments:

Line 47: please cite cite Tzedakis, P. C., et al. "Interglacial diversity." Nature Geoscience 2.11 (2009): 751-755.

Line 74: please cite De Boer et al. (2015) PLISMIP-ANT paper on which Dolan et al (2018) is largely based.

Line 74: please correct with "agree with how ANTARCTIC surface air temperature evolved"

Lines 79-82: I strongly disagree with this paragraph. Lost of long-term transient simulations have been performed, including MIS11, and you cite all those contributions in your introduction. I think what you mean is that no study really tried to improve the current simulations of MIS11-AIS, neither with climate forcing or ice sheet modeling, in absence of geological constraints on both climate and ice dynamics. Please reformulate this way, this much more honest. State that your aim is to improve by exploring aspects on which nobody really focused on so far (e.g. the two questions you pose at the end of this paragraph).

Line 89: correct as follows "of uncertainties in sea level reconstruction, and of uncertainties of the geometry…"

Figure 1: If the starting AIS topography is present-day (BEDMAP2 or other) please state it in this caption as well.

Figure 1: Are you sure that the glacial tongue in the Wilkes Land corresponds to Ninnis Glacier and not to Mertz Glacier instead?

Lines 101-102: please invert the order of the two sentences (put together everything about ocean forcing and then put the rest).

Line 105: "i.e., apply a transient surface temperature signal from the EDC ice core (Jouzel et al., 2007)". But Jouzel et al. only provide a temperature anomaly, what is your baseline climate forcing here for this thermal spin-up tase and then for the 5,000 kyrs geometry adjustment afterward?

Line 107: geometry is that of present-day, please specify which one and cite the reference (BEDMAP2, ALBMAP…).

Line 107: "We then let the AIS freely adjust for 5 kyr, between 425 and 420 ka": what is the ocean forcing for this 5,000 years free run? It seems to me that the topography

shown in Fig1 is really present-day. Is this really the AIS topography that you obtain after those 5,000 years of geometry evolution?

Line105-107: Please detail ALL the forcing, boundary conditions (geothermal heat fluxes, etc..) used for the entire spin-up procedure this 5000 years (even in the supplementary if you prefer). All experiments presented here, including the spin-up, must be reproductible.

Table 1: Do you really use only one enhancement factor (the same for both SIA and SSA)? If yes please indicate it within the Table.

Table 1: what about calving? How is this done?

Table 1: Why is the relaxation time set at 1 ayr while characteristic time is 3 kyr? Please provide a detail description in the supplementary about the choice of your parameters. Also provide a description of the sliding law, surface mass balance in the Supplementary.

Table 1: Units for salinity is "PSU", please fill the missing units.

Table 1: Please explain in the Supplementary how you choose the value for the thermal mixing coefficient (it varies quite a lot and this one of the main important parameter of oceanic parameterisation)

Table 2: Please substitute "Age scale" with "Age model".

Table 2: please provide a more detailed caption for this Table. What does "Age (ka)" corresponds to?

Table 2: Add a column to state what is the nature of the record (either dO18 or dD and it record is glaciological or marine).

Subsection 2.2: In this paper your focus is on MIS11. Can you explain why you chose to scale the ice cores isotopic records to the difference between LGM and PI? Thus because of this, how much do your glacial index scaled surface temperature differs

from the temperature form ice core records at DF, EDC and Vostok? I would like to see a Figure showing the derived surface air temperature for each GI and in comparison with each temperature reconstructions from dD for each ice cores used in this study.

Lines 123-124: I don't understand the choice of CCSM3 since many other runs form CCSM4, even earlier versions of CESM, were released by Otto-Bliesner's group for contribution to PMIP3 on CMIP5 platform for both PI and LGM, run by NCAR, on the same computer. CCSM4 presents strong improvements relative to CCSM3. I would like to see a discussion about this and related literature for both version CCSM3 and CCSM4 in the Supplementary.

Lines 126-127: On the contrary, I would like to see a few panels about simulated Antarctic climate and associated biases since it is also highly important to your study. Thus I am expecting you to also provide a bias correction to your forcing field (assuming the bias correction propagates linearity back in time). This is something that you did not do, but it needs to be done. I also expect to see a figure of surface air temperature over Antarctica and comparison with all available ice core records for LGM (not only the few that you consider here), to have a comprehensive view of the performance of your climate forcing.

Line 134: Please substitute "age scale" with "age model".

Line 140-141: I don't understand how you can compare dO18 from marine sediments and dD from ice cores and deduce that Holocene temperature history is inconsistent between those two. First of all, it is not straightforward to compare marine and glacial records togethers. To me this figure 2a does not make any sense, remove it.

Subsection 2.3.2: Please refer to Figure 7 to show the filtered GI.

Subsection 2.3.3: I find the choice of your sea level curve a bit awkward. Why not considering also Waelbroeck et al (2002) which also encompasses MIS11 and which is one of the best curve we have with Bintanja et al. (2008).? Actually, many other new

isotopic reconstructions have been done (e.g. Sutter et al., 2019), which is performed with more recent versions of models that Bintanja. Please redo some simulations also considering at least Waelbroeck et al (2002) in your ensemble.

Subsection 2.3.4: The methodology to provide intermediate geometries is definitely highly science-fiction. One can provide geometries, even though idealised, but with a more appropriate approach. For example: you could have done an equilibrium simulation with LGM conditions scaled with your GI for a few tens of thousands of year, and then transiently vary your climate forcing as in your control experiment until beginning of MIS11. This is a much better alternative than what you propose here. Or, alternatively, you can start one glacial cycle ahead and transiently vary your climate forcing with your GI. Then you could have used your the various GI generated with your scaling ensemble to vary the slope of transition from glacial to MIS 11. I strongly suggest you to try this way since, at least, you can justify much better your geometry ensemble than how you defined it currently.

Line 194: I think that the Figure number is wrong, it should not be Fig. 6.

Line 191-204: I am not sure in which Figure I can see the corresponding GI. Figure 4? If yes, I don't understand why you say that LR04-GI does not warm above PI temperature. PI temperature in Fig4a is given by 0 (the dashed horizontal line) right? To me LR04-GI goes beyond, even if not a lot.

Figure 4: Please put a horizontal dashed line corresponding to present-day AIS volume (26.9 x10ˆ6km3).

Lines 205-213: Actually, the amplitude of T° increase between al curve is broadly the same, as shown on your Figure 4a. The difference resides in the fact that LR04-GI starts with colder conditions that the others. However the ice volume evolution also decreases for a long time event with LR04-PI, however, because initially it the AIS grows, then it can not retreat beyond present-day extent during the peak of MIS11c. Vostok-GI yields a decrease in ice volume of the same order than LR04-GI, about

2x10ˆ6 km3.

Line 218: What I see on Fig 4b is that there is a tipping point, a threshold from which
the AIS retreats very fast. Thus, instead of warming rates, I see that when temperature
reaches a certain threshold, the ice sheet reacts fast. For example, the Vostok curve is
initially the warmest and thus the initial crease in volume is the strongest. Then the GI
stabilises compared to the other and the volume decreases slow down. I would thus
reformulate the analysis more in terms of tipping points and thresholds.

Line 224: What about surface melt? Do you have any in your simulations? What
method is used to calculate surface melt? Please provide detail about it in the Supple-
mentary.

Line 223-228: Could you provide a figure.

Figure 5: I would be nice to have a contour for SMB= 0m/yr, so to understand which
area are subject to surface melt.

Figure S2: you can't comparison between PI climate and ERA5 fields. . .this makes no
sense. Please modify this figure and show a comparison between present-day CCSM3
fields and ERA5 instead. Same for basal melting comparison: you can't use PI fields
and compare with present-day inferred basal melt rates from Rignot et al. By the way,
Which reference did you used in c) for basal melt rates?

Line 241-244: Thus why did you use an average over the last 10 kyrs. . .this does not
make sense, because orbitals are varying so much. Please remove the corresponding
results from the manuscript.

Line 249: I disagree. Trajectories are the same, they are only delayed, please reformu-
late.

Line 254: "This effect seems to be non-physical, and a result of the delay introduced
by the low-pass filter. " —-> The effect is physical, this is the result of your delayed
curve. Please remove this sentence. Because it is not the point here.

Line 256-259: "The 1 kyr low-pass GI is the only one that still preserves some higher-frequency variability ". I don't this on the Figure, I disagree. None of the filtered curve preserve the high frequency visible on the original EDC record.

Subsection 3.5: the only difference visible is before the threshold at 412k for EDC and DF index. This is because there is this threshold that initial geometry does not impact on your results. Basically, ocean forcing is driving all your scenarios. To see the difference in initial ice sheet geometry, you should turn-off the ocean forcing. But this wouldn't make sense. So the conclusion here is that ocean forcing is driving the initial trajectories until 412k, the tipping point. Thus is it not surprising that initial geometry does not matter too much. There is one thing you have not tested though here, is the variation in ocean forcing. Those tests makes also a lot of sens because ocean forcing has this tremendous effect on your simulations. Thus I would like to see a couple of other transient simulations with lower ocean forcing. And thus, try again your geometry scenarios with the difference ocean forcing rather than with EDC-GI or DF-GI.

I also would like to see a figure in the supplementary showing the Tforcing for each GI.

Figure 12: Please also add total AIS sea level contribution on the figure for each geometry.

Line 321-323: Please show some calving fluxes against oceanic warmth because you never really discuss calving, neither describe the calving method used here. Put this in the supplementary.

Line 340-353: I really would like to see a specific figure in the Supplementary of temperature forcing derived from GI for each ice core records and compared with the temperature reconstructed from dD of those ice cores.

Line 349-350: I completely disagree with this statement. On your Figure 1, you can definitely see that this is because EDC-GI and DF-GI yield temperature warmer than those I Vostok and thus it is a matter of tipping point rather than duration. . .

Line 391-392: "WAIS collapse was caused by the duration rather than the intensity of warming ". I don't see how you can conclude this here. I find the entire set of simulations rather inconclusive for this aspect. There is a tipping point in all the simulations shown in Figure 4. However, the amplitude of contribution to sea level is determined then by the magnitude of the warmth during the peak rather than the duration of the peak itself. Actually the ice sheet retreat in a very comparable way when using EDC and DF, which have a different peak duration...IN fact there is almost no significant difference between them in Figure 12 as well.

Line 395-400: Instead of just stating it, show it. Plot air and ocean temperature forcing versus WAIS ice volume evolution; same for EAIS.

---

## Author Comment (AC1) · 31 Aug 2020

Dear editor, dear Dr. Schanwell,

We thank the reviewer for his constructive, insightful and helpful evaluation which we feel helped to improve the manuscript. This instigated additional modeling that resulted in numerous refinements, and significant upgrades to the model description and discussion sections. Below, we provide a point-by-point response to each comment, which we numbered in red for easier reference. Our response is structured as follows: Referee comment (*in black italics*), author's response (in green), and proposed changes in the original manuscript text (*in blue italics*) where significant rewriting was done to include the suggested changes. We also add to the end of each figure caption (in blue) their proposed numbering in the revised version of the manuscript.

**Main concerns**

**1.** *"[…] I suggest to expand section 2.1 to add this required information. To be more specific, what type of stress balance does SICOPOLIS use? What kind of basal friction law do you apply? I know you list the parameters in Table 1, but without the corresponding equation, they are rather useless. Does your basal friction coefficient vary spatially and/or temporally? What are your boundary conditions for your enthalpy equation (e.g. do you specifiy a geothermal heat flux? Is it spatially constant?)? How do you treat calving in the model? There are a number of ways how to parameterise this. Since you talk about this in your results, it is essential to know how this is handled in your model.*
*Also you should mention with what geometry your initialise your model. I believe it is with present-day geometry, but with which dataset (Bedmap2, Bedmachine)? Do you take the bedrock and ocean floor topography from the same dataset?"*

**Our response:** We have expanded the model description, including the requested additional information, as can be seen below:

*"For our experiments we employ the 3D thermomechanical polythermal ice-sheet model SICOPOLIS (Greve, 1997; Sato and Greve, 2012) with a 20 km horizontal grid resolution and 81 terrain-following layers in the vertical. It uses the one-layer enthalpy scheme introduced in Greve and Blatter (2016), which is able to correctly track the position of the cold-temperature transition in the thermal structure of a polythermal ice body. The model combines the Shallow Ice (SIA) and Shallow Shelf (SSA) approximations using*

$$U = (1 - w) \cdot u_{sia} + u_{ssa}$$

*where U is the resulting hybrid velocity, $u_{ssa}$ and $u_{sia}$ are the SSA and SIA horizontal velocities, respectively, and w is a weight computed as*

$$w = \frac{2}{\pi} \arctan\left(\frac{u_{ssa}^2}{u_{ref}^2}\right)$$

*where the reference velocity, $u_{ref}$, is set to 30 $ma^{-1}$, which reduces the contribution from SIA velocities mostly in coastal areas of fast ice flow where this approximation becomes invalid. Basal sliding is implemented within the computation of SSA velocities as a Weertman-type law (cf. Bernales et al., 2017a, Eqs. 2–6). Sliding coefficients are adjusted during the equilibrium calibration run such that grounded ice thickness matches the present-day observations from the Bedmap2 data set (Fretwell et al., 2013) as close as possible. This adjustment process follows the iterative method of Pollard & DeConto (2012b), where the coefficients are allowed to vary spatially, but not temporally, outside of the adjustment phase. Sliding coefficients in sub-ice shelf and ocean areas are set to $10^5$ m $yr^{-1}Pa^{-1}$, representing soft, deformable sediment, in case the grounded ice advances over this region. The initial bedrock, ice base, and ocean floor elevations are also taken from Bedmap2. Enhancement factors for both grounded and floating ice are set to 1, based on sensitivity tests in Bernales et al.*

*(2017b). This choice provides the best match between observed and modelled ice thickness, similar to the findings in Pollard and DeConto (2012a).*

*Surface mass balance is calculated as the difference between accumulation and surface melting. The latter is computed using a semi-analytical solution of the positive degree day (PDD) model as in Calov and Greve (2005). Near-surface air temperatures entering the PDD scheme are adjusted through a lapse rate correction of 8.0 °C km⁻¹ to account for differences between the modelled ice sheet topography and that used in the climate model from which the air temperatures are taken. For the basal mass balance of ice shelves, we use a calibration scheme of basal melting rates developed by Bernales et al. (2017b) to optimise a parameterisation based on Beckmann and Goosse (2003) and Martin et al. (2011) that assumes a quadratic dependence on ocean thermal forcing (Holland et al., 2008; Pollard and DeConto, 2012a; Favier et al., 2019). This optimised parameterisation is able to respond to the variations in the Glacial Index (GI, Sect. 2.2). A more detailed description of this parameterisation is given in the supplementary material. In our experiments, we prescribe a temporal lag of 300 years for the ocean response to GI variations, which is considered the most likely lag in response time of the ocean compared to the atmosphere in this region (Yang and Zhu, 2011). At the grounding line, the basal mass balance of partially floating grid cells is computed as the average melting of the surrounding, fully floating cells, multiplied by a factor between 0 and 1 that depends on the fraction of the cell that is floating. This fraction is computed using an estimate of the sub-grid grounding line position based on an interpolation of the current, modelled bedrock and ice-shelf basal topographies. At the ice shelf fronts, calving events are parameterised through a simple thickness threshold, where ice thinner than 50 m is instantly calved out.*

*Glacial isostatic adjustment is implemented using a simple elastic lithosphere, relaxing asthenosphere (ELRA) model, with a time lag of 1 kyr and flexural rigidity of $2.0 \times 10^{25}$ Nm, which was found by Konrad et al. (2014) to best reproduce the results of a fully-coupled ice sheet–self-gravitating viscoelastic solid Earth model. The geothermal heat flux applied at the base of the lithosphere is taken from Maule et al. (2005) and is kept constant throughout the simulations. All relevant parameters used in the modelling experiments are listed in Table 1."*

*"**2.** Could you please motivate the ensembles or parameters changes that you are investigating a bit more? As it stands now, it seems like you picked a number of parameters, but there also could an argument be made for a bunch of other parameters to be varied."*

**Our response:** We picked these ensembles as they are inherent sources of uncertainty that were not addressed by any previous studies that included MIS11. We also performed additional tests to support our parameter choices (such as ocean temperature, lag in its response, and the choice of climate model), which we included in the supplementary material. We justify the choice of ensembles in the last paragraph of the introduction:

*"For this purpose, we perform five ensembles of numerical simulations of the AIS and focus on aspects that remain unaddressed by previous studies. We evaluate the impact on resulting ice volume and extent of the choice of proxy records (including their differences in signal intensity and structure), of the choice of sea level reconstruction, and of uncertainties in assumptions regarding the geometry of the AIS at the start of MIS11c. Additionally, we assessed the robustness of our results to the uncertainty in external climate forcings, which we present in the supplementary material."*

*"**3.** I find most of the figures (e.g. 4, 6,7,9) not very informative. Looking at integrated quantities is OK, but having five Figures like that is too much. I suggest to combine them into a Figure with several panels. I also find it hard to judge in these volume plots whether differences are small or large (Is 2000 km3 a lot?). Maybe better to plot it in percent normalised to your starting volume? Also just because your ice volume is similar does not mean you cannot have regional differences in grounding-line position or ice thickness. For example on P16L279 you state ". . . show similar retreat rates..." but I cannot find a Figure where this is actually shown. So I suggest to add some Figures, where we can also look at some spatial differences (a few suggestion in the technical corrections below). For example, you could plot some grounding-line positions from different simulations in 2D on top of each other to see the differences in retreat or lack thereof. I also encourage the authors to discuss their results more in depth. For example, they state in L276ff that different initial ice sheet configurations converge to the same geometry for the same climate forcings. This alone is quite surprising to me and at least warrants a discussion why potential feedback mechanisms (e.g. stabilising grounding-line on topographic height) are not triggered in these simulations?"*

**Our response:** We have added a compilation of all different GIs as a panel in Fig. 2, which allows us to remove them from Fig. 1 (see Figs. 1 and 2 below). Regarding figures 4, 6, 7, and 9, they were restructured, with their (b) panels being merged into a single figure, and added a line that represents present AIS volume, as suggested by Reviewer 2 (see Fig. 3 below). We have added a new figure (see Fig. 4 below) where we show the grounding lines for each of our core experiments at times of interest: 420, 415, 410, and 405 ka. Regarding the fact that different configurations converge to the same geometry, we have found a tipping point at 412 ka (as pointed out by Reviewer 2 in his comment 47), where the ocean forcing under the main ice shelves (cf. thermal forcing in Fig. 8) is strong enough to drive ice sheet retreat in all geometry scenarios. There are two grounding-line stabilising feedbacks not included in our current version of the model: (*i*) a local sea-level drop caused by a reduced gravitational attraction of a shrinking ice sheet (e.g. Mitrovica et al., 2009), and (*ii*) the observed faster rebound of the crust due to a lower mantle viscosity in some WAIS locations (Barletta et al., 2018). Even though our ELRA model is set up with a relatively fast response time of 1 kyr (compared to the standard 3 kyr), the resulting bedrock uplift is still not able to trigger a stabilizing effect that compensates for the strong ocean-driven retreat. These feedback mechanisms during MIS11c could be further investigated through the utilization of an Earth-ice coupled model, which is certainly an interesting topic for future research. These points will be incorporated in our Discussion section.

[Figure]

Figure 1: Surface topography of the AIS at the start of our core experiments (425 ka), based on a calibration against Bedmap2 (Fretwell et al., 2013; see Sect. 2.1). The locations mentioned in the text, including the drilling sites of the ice (circles) and sediment (red diamonds) cores on and around Antarctica, are showcased. This is Fig. 1 after the revisions to the manuscript.

[Figure]

Figure 2: Reconstructions used in this study: (a) LR04 δ¹⁸O (black) and ice-core δD [‰]; (b) resulting Glacial Indices from the reconstructions in (a) (cf. Sect. 2 and Table 2); (c) global mean sea level anomaly relative to PI (meter sea level equivalent, m s.l.e.). This is Fig. 2 after the revisions to the manuscript.

Figure 3: Sensitivity of AIS response (in total ice volume, $10^3$ km$^3$) between 420 ka and 394 ka to (a) CFEN GI reconstructions; (b) SSEN rescaled GI reconstructions; (c) RSEN low-pass filtered GI reconstructions; (d) SLEN sea level reconstructions forced by EDC GI (cf. Table 4). This is Fig. 4 after the revisions to the manuscript.

[Figure]

Figure 4: Grounding lines at 420, 415, 410, and 405 ka for the CFEN simulations. This is Fig. 5 after the revisions to the manuscript.

*"**4.** I think you should scratch your attempt to identify drivers for future change. You have it in your research questions, but other than in the conclusion section you never mention it again. And your statement in the conclusion statement is extremely vague (and we know this already) and to be honest not backed up by your simulation results."*

**Our response:** We agree with the reviewer regarding the relative weakness of this section, and have removed the mention of drivers for future change from our research questions and the conclusions.

*"**5.** The abstract in its current form is much too long and too descriptive. Please shorten and make more concise."*

**Our response:** we have shortened the abstract, and modified it to also account for the analyses suggested by Reviewer 2. It now reads:

*"Studying the response of the Antarctic ice sheets to periods when climate conditions were similar to the present can provide important insights for understanding the observed changes and help identify natural drivers of ice sheet retreat. The Marine Isotope Substage 11c (MIS11c) interglacial is one of the best candidates for an in-depth analysis, given that during its later portion, orbital parameters were close to our current interglacial. Although Antarctic ice core data indicate that MIS11c $CO_2$ levels were close to Pre Industrial, they also show that warmer-than-present temperatures (of about 2 ºC) lasted for much longer than during other interglacials. While substantial work has been conducted regarding the response of the Greenland Ice Sheet to MIS11c climate, the response of the Antarctic ice sheets and their contribution to sea level rise remain unclear. We improve the current constraints for this period using a numerical ice-sheet model forced by MIS11c climate conditions derived from climate model outputs scaled by three glaciological and one sedimentary proxy records of ice volume. Our results suggest that the East and West Antarctic ice sheets contributed with 6.7 to 8.2 m to the MIS11c sea level rise, independently of the choice of sea level reconstructions and multi-centennial climate variability. The main source of uncertainty arises from the sensitivity of the East Antarctic Ice Sheet response to the initial configuration. We have found that the regional climate signal of the MIS11c peak warming in Antarctica captured by the ice core records is necessary for the recorded sea level highstand to be reproduced, and that oceanic warming of only 1.5 ºC under the Ross and Filchner-Ronne ice shelves, if sustained for ca. 4 thousand years, leads to a West Antarctic Ice Sheet collapse."*

*"**6.** This is more an optional point and maybe a matter of taste, but I think you could also add a model limitations section. There are a few places where you can shorten the main text (see below), so that this would not much increase the length of the manuscript. I always find it helpful in modelling papers to have a section in which limitations and potential future avenues for improvements are discussed. I must admit that as the paper stands now with very little information about the ice-sheet model, it is hard to examine what the benefit of your model setup is?"*

**Our response:** We expect that the changes made in the methods (included as a response to comment 1) help to partly clarify this issue. We have further opted to discuss the model limitations within the context where they were relevant in the discussions, as opposed to giving them their own section.

**Technical corrections**

**7.** *"L8 I do not think that the Greenland information is necessary in the abstract. Also the latter half of the sentence makes no sense to me ". . . , both configurations of the Antarctic ice sheets. . . "? What configurations?"*

**Our response:** We have removed the mention of the Greenland Ice Sheet sea-level contribution from the abstract, and we expect that the reformulation presented above (comment 5) has clarified the text.

**8.** *"L12 Does LR04 need to be introduced as an acronym? I did not know straight away what it is."*

**Our response:** We have removed the LR04 acronym from the abstract while making it more concise and less descriptive as requested (see response to comment 5).

**9.** *L17 Here and throughout, I find the term "ice-sheet contraction" unusual. I know what you mean, but I think more commonly it is referred to as "ice-sheet retreat". Please consider changing it.*
**Our response:** We had used "contraction" since the changes seen are both in extent and volume. Nevertheless, we reverted to the usual term as suggested.

**10.** *L29-34 This sentence is way too long and confusing. Please split up and make clearer.*
**Our response:** We recognise the sentence was indeed too long, and have rewritten the passage. It now reads:
*"However, Dutton et al. (2015) point that climate modelling experiments with realistic orbital and greenhouse gas forcings fail to fully capture this MIS11c warming despite the fact that orbital parameters were almost identical to Present Day (PD) during its late stage (EPICA, 2004; Raynaud et al., 2005). Earlier studies (e.g., Milker et al., 2013; Kleinen et al., 2014) have also shown that climate models tend to underestimate climate variations during this interglacial, for which ice core reconstructions show the mean annual atmospheric temperatures over Antarctica to have been ca. 2 °C warmer than Pre-Industrial (PI) temperatures."*

**11.** *L43 What do you mean by "reduced stability"? And why would that trigger stronger glacial-interglacial cycles?*
**Our response:** Holden et al. (2011) show that a reduced stability of the WAIS (i.e., a higher susceptibility to collapse) through time is caused by an increased bedrock relief as a result of continuous erosion, while Holden et al. (2010) show that the positive feedback of a strong WAIS retreat could contribute to these stronger cycles. We have added the Holden et al. (2010) citation and rewrote this part of the introduction as shown below. We refrained from discussing the mechanisms for the mentioned feedback in detail, since they are not the focus of our work.
*"The length of this unusual interglacial and the transition to stronger glacial-interglacial cycles seen in the recent geological record may have been triggered by a reduced stability of the West Antarctic Ice Sheet due to cumulative bed erosion (WAIS, Fig. 1; Holden et al., 2011), which in turn provided a positive climate feedback (Holden et al., 2010)."*

**12.** *L49-55 I think this paragraph can be thrown out, as it is irrelevant to the Antarctic simulations in the paper. It suffices to say, I believe, that the ice-sheet history in Antarctica is more uncertain than for Greenland.*
**Our response:** We have removed the paragraph, and moved the appropriate references to the beginning of the next paragraph, which we start with
*"The MIS11c history of Antarctica is less constrained than that of Greenland (e.g., Willerslev et al., 2007; Reyes et al, 2014; Dutton et al,. 2015; Robinson et al, 2017)."*

**13.** *L56-58 The first half of the sentence is confusing. The way it is written, it makes it sound as if Raymo and Mitrovica estimated it to be 6-13 m above present-day? But why is there a reference to Dutton et al. then? Here and throughout, could you please try to keep sentences shorter. It makes it easier to follow for the reader.*
**Our response:** We have removed the Dutton et al. reference, as it was misplaced there. We tried to rewrite the sentences to be shorter and easier to follow where necessary.

**14.** *L61-64 Again a very long sentence which I do not understand. Please break up the sentence and clarify.*
**Our response:** We have rephrased the sentence:
*"Counter-intuitively, the dating of moraines in the Dry Valleys to MIS11c is used to indirectly support regional ice sheet retreat (Swanger et al., 2017). Swanger et al. (2017) argue that ice sheet retreat in this region would result in nearby open-water conditions and thus a source of moisture and enhanced precipitation, fueling the local glacier growth that produced the moraines in the Dry Valleys."*

**15.** *L65-80 Here, I would like to see what your study adds to studies like the one from Sutter et al. 2019. What is the advantage of your study/model setup ?*

**Our response:** Our study has different objectives than that of Sutter et al. (2019), and thus uses different approaches. For example, we focus on MIS11 exclusively, evaluating different transient climate signals and testing for a different set of factors that can influence ice sheet simulations. We made this clearer by rewriting the last two paragraphs of the introduction:

*"As detailed, many modelling studies have investigated AIS responses over time periods that include MIS11. However, so far none has focused specifically on this period. Given a dearth of information for MIS11 and conflicting constraints on how Antarctica responded to this exceptionally long interglacial (Milker et al., 2013; Dutton et al., 2015), we here focus on MIS11c, the peak warming period between 420 and 394 ka. Our aim is to reduce the current uncertainties in the AIS behaviour during MIS11c, specifically addressing the following questions:*
*[...]*
*For this purpose, we perform five ensembles of numerical simulations of the AIS and focus on aspects that remain unaddressed by previous studies. We evaluate the impact on resulting ice volume and extent of the choice of proxy records (including their differences in signal intensity and structure), of the choice of sea level reconstruction, and of uncertainties in assumptions regarding the geometry of the AIS at the start of MIS11c. Additionally, we assessed the robustness of our results to the uncertainty in external climate forcings, which we present in the supplementary material."*

**16.** *L81 I do not agree that you are presenting model reconstructions. What you present are sensitivity experiments. But as far as I can tell, you are not trying to match any geological constraints which is what I understand as model reconstruction.*

**Our response:** Geological constraints are very scarce for this period, and we discuss how our simulations match the available constraints throughout the manuscript (e.g., L313-318 and L368-375 in the original submission). Nevertheless, it is a good point that our experiments can be seen as sensitivity experiments. For this reason, we have refrained from using the term and rewrote this sentence also with input from Reviewer 2:

*"Given a dearth of information for MIS11 and conflicting constraints on how Antarctica responded to this exceptionally long interglacial (Milker et al., 2013; Dutton et al., 2015), we here focus on MIS11c, the peak warming period between 420 and 394 ka. Our aim is to reduce the current uncertainties in the AIS behaviour during MIS11c, specifically addressing the following questions: [...]"*

**17.** *L85 As said above, I do not think you really address the last question about future ice-sheet changes. Therefore, I recommend removing it from the manuscript altogether.*

**Our response:** We have removed this question.

**18.** *L106 In addition to the changes suggested above. How do you initialise for the different ice-sheet configurations? Do you use the same temperature spin-up and let it evolve afterwards? Or do you let it evolve to a different geometry and do the temperature spin-up then with a fixed geometry?*

**Our response:** In response to your comments and those of Reviewer 2 (comment 33) regarding our different geometries, we have changed our approach. We force the thermally spun up ice sheet with LGM conditions for 5 kyr so it grows to an intermediate stage between PI and LGM extent, and then place the resulting geometry at different points in time: 420, 425 and 430 ka. We then let these transiently evolve from then until 395 ka, and analyse the period between 420 and 395 ka, as in the original submission. We made changes to the text (see below) and to Table 3 to reflect this new approach:

*"In order to create a representative range of initial geometries at 420 ka, we use a common starting geometry, but vary the relaxation time (0, 5, and 10 kyr). For this common starting geometry, we perturb the thermally spun up AIS with a constant LGM climate (i.e., temperature and precipitation) without sub ice-shelf melting, allowing it to grow to an intermediate extent between PI and LGM over a 5 kyr period. We assume this to be the starting AIS geometry at 420 (Fig. 3a), 425, and 430 ka, and let it transiently evolve from then. Table 3 summarises the procedure to create each initial ice sheet geometry (labelled gmt1 to gmt3; Fig. 3a-c). The gmt1 initial topography is generally more extensive and thinner*

*than the control. Its grounding line advanced at the southern margin of the Filcher-Ronne Ice Shelf and at Siple Coast, but the ice sheet interior is on average 200 m thinner than the control and indeed up to 500 m thinner across particular regions such as the dome areas of the WAIS and Wilkes Land (Dome C). It is, however, about 200 m thicker at its fringes, which results in a gentler surface gradient towards the ice sheet margins. The gmt2 initial topography is less than 100 m thinner than control over the EAIS interior, and ca. 100 m thicker over the WAIS interior and at the EAIS margins. Finally, the gmt3 initial topography is overall thicker than control, though not by more than 100 m except at the western side of the Antarctic Peninsula and the WAIS margins, where some regions are up to 300 m thicker (Fig. 3)."*

**19.** *L106 From where do you get your surface temperature distribution? An ice core only provides you with temperature changes with respect to a certain baseline. Please add this to this section.*
**Our response:** Based on this comment and on comment 20, which led us to slightly alter our approach (as detailed in the answer to comment 20), we have rewritten the paragraph for increased clarity:
*"All ensembles cover a period from 420 to 394 ka. To initialise the AIS, we first perform a thermal spin-up over a period of 195 kyr from 620 to 425 ka, i.e., apply a transient surface temperature signal from the EDC ice core (Jouzel et al., 2007) as an anomaly to our PI climate (described in the next section) while keeping the ice sheet geometry constant at our previously calibrated Bedmap2-based configuration. We then let the AIS freely evolve for 5 kyr, between 425 and 420 ka, applying transient GI forcing during the relaxation period. We chose 425 ka as the starting point for relaxation because it is when the oxygen isotope value in the EDC ice core is closest to PI during our study period. In summary, we ignore the first 5 kyr of our simulations to avoid a shock from suddenly letting the ice-sheet topography freely evolve at the start of our period of interest."*

**20.** *L107-109 This means you just move this shock outside of your time period of interest? This is in general OK, but raises the following questions: What forcing do you apply for the 5 ka in which the ice geometry is allowed to freely evolve? And how far away do you get from your initial geometry? And I am also missing a plot where you show that your ice sheet is close to steady state. I would appreciate if you could add a plot for this.*
**Our response:** We had initially applied the same GI forcing based on the EDC core for all simulations during the relaxation stage, so that they all had the same geometry at 420 ka. This proved to be a problem for the LR04-forced simulation, since it significantly deviates in its isotope values from the others. Consequently, and in response to the review, we now apply the GI forcing during the relaxation stage that corresponds to the forcing during the main experiments (i.e., the 425-420 ka DF GI for the DF-forced runs, EDC GI for the EDC-forced runs, and so on). A figure showing the spread in initial geometries during this period is now provided in the supplement (see Fig. 5 below). We do not provide a plot showing that the ice sheet is close to steady state because the point of the thermal spin-up is precisely to remove the effects of the initial steady state (attained during the calibration of the model) from our simulations, and offer a more realistic internal thermal structure for the AIS. All figures shown already contain the new simulations, and the corresponding part in the Methods section is also revised as shown in comment 19.

[Figure]

Figure 5: Relaxation period between 425 and 420 ka for all four CFEN members. (a) shows the grounding line at 420 ka (solid line) and at 425 ka for each member (dashed lines); (b) shows the evolution of total ice volume [$10^6$ km$^3$] during this 5 kyr period for each member. Dashed line shows the volume of the present-day AIS according to Bedmap2. This is Fig. S12 after the revisions to the supplement.

**21.** *L129, equation (2): From this equation I gather that you apply the same temperature differences to the ocean as you do to the atmosphere? And you also do not apply a time lag to the ocean warming/cooling? Is that really realistic giving the long response time of the ocean compared to the atmosphere? At the very least, this choice should be discussed somewhere in the text.*

**Our response:** We do not apply the same temperature differences to ocean and atmosphere, but modulate them with the same index. The differences are obtained by the ocean temperature and atmospheric temperature fields from the climate forcing. We appreciate this criticism regarding the ocean lag, also voiced by Reviewer 2 (in his comment **4**), and have acted accordingly. We have introduced a lag to the ocean forcing of 300 years, as this is the timescale of response of the Southern Ocean (Yang & Zhu, 2011). We additionally present in the supplement an ensemble of sensitivity tests to different time lags in the ocean forcing (see Fig. 6 below), which shows their effect to be very small compared to the timescales of this study. We tried to clarify the concern about the differences applied by rewriting the last sentence before Eq. (2):

*"The atmospheric and ocean temperature (T) fields at time t are reconstructed based on their respective PI and LGM reference fields ($T_{PI}$ and $T_{LGM}$ respectively) using: [...]"*

[Figure]

Figure 6: Sensitivity of the AIS response expressed in total ice volume [$10^6$ km$^3$] to a range of lags (0-1600 years) between the atmospheric forcing and the ocean forcing between 420 and 394 ka. This is Fig. S10 after the revisions to the supplement.

**22.** *L137 To me all headers in this section should rather read "Model sensitivity to XXX". Because this is ultimately what you do in this paper, rather than rigorously quantifying uncertainties.*
**Our response:** We have changed it accordingly. We thank for this good suggestion as it also makes it consistent with the headers in section 3 (results).

**23.** *L139-141 This sentence needs rewriting. I am not sure I understand what you are saying.*
**Our response:** We have rewritten this section, also based on an additional request from Reviewer 2 (comment 30):
*"Because different approaches have been used to transform the isotope curves into a GI, we assess the sensitivity to the choice of the scaling procedure by performing an additional scaling using another reference value for $\delta X_{LGM}$. In the new scaling procedure, $\delta X_{LGM}$ is the average (between 19 ka and 26.5 ka) rather than the peak value. We compare the effects of using these two procedures when applied to the EDC ice core $\delta D$ and the LR04 stack $\delta^{18}O$ records. We call this ensemble the Scaling Sensitivity Ensemble (SSEN)."*

**24.** *L154 should be "mean sea level"*
**Our response:** We have corrected as requested.

**25.** *L166 Here I believe you say that you also initialise with present-day conditions, but this needs to come much earlier and with more info as to what datasets you used for this.*
**Our response:** We expect that the changes made to the Methods section as described above (comment 1) have successfully addressed this issue. Hence, in this paragraph we merely provide a reference to section 2.1:
*"Similar studies that assessed AIS changes over (one or more) glacial and interglacial cycles often adopt a PI or PD starting geometry (e.g., Sutter et al., 2019, Tigchelaar et al., 2019, Albrecht et al., 2020). We have followed the same approach in our CFEN experiments (see Sect. 2.1)"*

**26.** *L221 you state: ". . . ice sheet contraction is associated with strong basal melting close to the grounding lines ...". First of all this comes a bit out of the blue. Secondly, you show little evidence that this is actually the case. In Fig. 5 you show that basal melting is dominating, but if you have different SMB rates, the basal melt rate could be either 1.5m/yr or 6 m/yr. Please also avoid relative terms like "strong" without giving any numbers. Do you mean 5, 50, or 500 m/yr when you say "strong" melting. Related to this, do you apply melting to partially grounded grid cells or only to fully floating? This makes a big difference how much your grounding line retreats for similar melt rates.*

**Our response:** Based on this and other comments from the reviewers, we have moved this paragraph to the Discussion section, where it is more fitting and does not "come out of the blue". We added the information about the basal melting to the Methods section (shown above in our response to comment 1). Also, we expect that changes made to Fig. 5 in the original manuscript (see Fig. 7 below) further help clarify the regions where SMB or ice-shelf basal melting dominates.

[Figure]

Figure 7: Surface mass balance (SMB, ma$^{-1}$) for the grounded ice and basal melting (Q$_{bm}$, ma$^{-1}$) for the ice shelves for the CFEN simulations at 415 ka. Hatched areas show where basal melting dominates over surface mass balance and where surface mass balance is positive (i.e., where surface ablation occurs). Everywhere where |Q$_{bm}$| > |SMB|, ice shelves are thinning. This is Fig. 8 after the revisions to the manuscript.

**27.** *L222 should read ". . . Siple Coast, at the Ross Ice Shelf, and underneath ..."*
**Our response:** We thank the reviewer for spotting the typo and have corrected it.

**28.** *L223-224 & L227 Since your basal melt rate is a quadratic function of your ocean temperature, stating that it is a combination of warming of the upper ocean layer and high melt rates is saying the same thing. Please reformulate.*
**Our response:** This sentence has been removed, and we focus our discussion on the thermal forcing under the ice shelves, as opposed to the distribution of SMB vs. ice-shelf basal melting (Figs. 7, and 8).

**29.** *L228 Two things here. First, since you have a separate results and discussion section, I was expecting only a description of the results. However, here and in other places (e.g. L245, L256-259) in your results section you are interpreting and discussing your results already. So either you have a combined results and discussion section or you move this material to your discussion section. Secondly, I cannot confirm your statement that ice loss is dominated by surface ablation on Amery in Fig.5. First of all, the panels are too small, so I am not sure if Amery is hatched or not? I do not really understand the purpose of Fig. 5, but to me Amery looks pretty red which means to me that there is a lot of ablation in this area. So why would it not retreat there and why is ablation so high in this region compared to basal melting?*
**Our response:** Thank you for highlighting these points. We have moved this part to the discussion as suggested. As described in comment 26, Fig. 7 in this response letter shows significant improvements related to Fig. 5 in the original manuscript, and now better highlights the regions affected by basal melting and surface ablation. In combination with a new figure provided (see Fig. 8 below), we were able to see that Amery is indeed, contrary to what we originally stated, dominated by basal melting. However, the difference between surface ablation and basal melting is not as pronounced as in the larger ice shelves, such as Ross and Filchner-Ronne. We will make the necessary adjustments to the text.

[Figure]

Figure 8: Evolution throughout MIS11 for each CFEN member averaged over specified Antarctic ice shelves for (a) Summer surface air temperature [ºC]; (b) thermal forcing under the ice shelves (i.e., ocean temperature minus ice base temperature, in ºC); and (c) sea level contribution by EAIS and WAIS. Colours denote the respective CFEN member, while line styles denote each ice shelf (panels a,b), and each ice sheet (panel c). DML refers to an average for ice shelves along the Dronning Maud Land between 27ºW and 30ºE. This is Fig. 9 after the revisions to the manuscript.

*30. L237 ". . . , the resulting ice sheet histories are quite similar." This is true for the integrated ice volume, but again I find this quite superficial and it could be different when we look at 2D fields.*
**Our response:** We have included a new figure (Fig. 4 above) to show the evolution of the grounding lines of each ensemble member at key times, and that further supports this statement that their histories are indeed fairly similar.

*31. L242 If it is problematic why did you include it?*
**Our response:** We have removed these from our study, as also requested by Reviewer 2.

*32. L256-259 This is discussion for me (see comment above).*
**Our response:** Indeed, we agree and have moved it to the discussion.

*33. L268-L274 This paragraph should rather be part of your experimental design section. By now there are so many simulations that you performed that I think it is really necessary to add a table where you list all the simulations with important forcing parameters in a table. It is really hard to keep track of the simulations.*
**Our response:** We agree with the reviewer; this paragraph felt out of place and was essentially recapping part of what we described in the methods. We have removed it. We appreciate the suggestion for a summary table, which we added to the end of the Methods section.

*Table 4. Summary of performed experiments grouped by ensemble, listing their respective GI forcings, used sea level reconstruction and choice of initial geometry. LGMavg denotes that the GI was rescaled using the average LGM value as opposed to the peak value (cf. Sect. 2.3.1 and Table 5). The SGEN experiments were grouped for better visualisation, but each SGEN row corresponds to 3 experiments, one starting from each geometry (1 to 3).*

| Ensemble | Experiment | GI forcing | Sea level reconstruction | Initial Geometry |
|---|---|---|---|---|
| CFEN | lr04 | LR04 | Bintanja and van de Wal (2008) | control |
| CFEN | edc | EDC | Bintanja and van de Wal (2008) | control |
| CFEN | edf | DF | Bintanja and van de Wal (2008) | control |
| CFEN | vos | Vostok | Bintanja and van de Wal (2008) | control |
| SSEN | lr04lgmavg | LR04$_{LGMavg}$ | Bintanja and van de Wal (2008) | control |
| SSEN | edclgmavg | EDC$_{LGMavg}$ | Bintanja and van de Wal (2008) | control |
| RSEN | lp1bx | EDC (1 kyr low pass) | Bintanja and van de Wal (2008) | control |
| RSEN | lp3bx | EDC (3 kyr low pass) | Bintanja and van de Wal (2008) | control |
| SLEN | lp5bx | EDC (5 kyr low pass) | Bintanja and van de Wal (2008) | control |
| SLEN | s16l | EDC | Spratt and Lisiecki (2016) long | control |
| SLEN | s16s | EDC | Spratt and Lisiecki (2016) short | control |
| SLEN | s16u | EDC | Spratt and Lisiecki (2016) upper uncertainty | control |
| SLEN | spm | EDC | Imbrie et al. (1989) | control |
| SLEN | wae | EDC | Waelbroeck et al. (2002) | control |
| SGSEN | edcgmt[1-3] | EDC | Bintanja and van de Wal (2008) | gmt1-3 |
| SGSEN | edfgmt[1-3] | DF | Bintanja and van de Wal (2008) | gmt1-3 |
| SGSEN | vosgmt[1-3] | Vostok | Bintanja and van de Wal (2008) | gmt1-3 |

**34.** *L276 To me that is really surprising. From my experience, the initial geometry is quite important with regard to what your results look like at the end of the simulation. You glance over this, but this needs a discussion. Why do you think this is the case?*

**Our response:** The insensitivity to the choice of initial geometry of the WAIS seems to stem from the fact that the ocean is able to trigger its collapse regardless of its initial state. The EAIS, for example, showed a clear sensitivity to the initial geometry. We have included this in our discussion:

*"The EAIS reacted sensitively to the choice of starting ice geometry, especially over Wilkes Land, where the spread in thickness and grounding line position is highest among its regions (Fig. 11). While drainage basins such as those of Totten and Dibble glaciers seemed to become more stable given a larger ice sheet, Cook glacier seems to thin regardless of the choice of initial geometry. Overall, the EAIS yields a contribution range of 2.4 to 3.7 m s.l.e. at 405 ka. Conversely, the WAIS was rather insensitive to the choice of geometry (yielding 4.3--4.5 m s.l.e. at 405 ka) due to the stronger role played by the ocean after 412 ka, which reduced its volume to a minimum in all simulations regardless of starting geometry."*

**35.** *L279 ". . . also show similar rates of retreat ...". Again this is nowhere shown. I mean in Fig. 10 it looks like they actually have exactly the same grounding-line position. Is that true?*

**Our response:** We modified Figs. 3 and 10 of the original manuscript (Figs. 9 and 10 presented below), also based on comment 43. The grounding lines are indeed close to each other, but are not at the same position. Also, by "rates" we mean their pacing, and not the starting and final volumes, which can be seen in Fig. 9 in the original submission. We have changed the phrasing in the text to avoid misunderstanding.

[Figure]

Figure 9: (a-c) Three different starting ice sheet geometries at 420 ka for the EDC CFEN member (gmt1-3). Color scheme shows differences in surface elevation between each geometry and the control for 420 ka (d). Differences are only shown where the ice is grounded in both geometries, and coloured lines show the respective grounding lines in gmt1-3, also overlain in (d). This is Fig. 3 after the revisions to the manuscript.

[Figure]

Figure 10: (a-c) ice sheet geometries at 405 ka for the EDC CFEN member using three different starting geometries at 420 ka (Fig. 12). Color scheme shows differences in surface elevation between each geometry and the control for 405 ka (d). Differences are only shown where the ice is grounded in both geometries, and coloured lines show the respective grounding lines in gmt1-3, also overlain in (d). This is Fig. 7 after the revisions to the manuscript.

**36.** *L282 Could you please add these locations to the respective Figure for better orientation.*
**Our response:** These locations were added to Fig. 1 (see our response to comment 3), so that it can be used as a reference for the locations cited in the manuscript, while the remaining figures can be less cluttered with text. Some glacier locations were also reviewed based on comment 12 from Reviewer 2, as we were originally pointing to adjacent glaciers instead.

**37.** *L289-301 This is a weak introduction to the discussion and repeats most of the material that you covered in the introduction. Consider removing it.*
**Our response:** We have removed it, while significantly reordering and rewriting most of the discussion, in light of the comments from both reviewers.

**38.** *L321 ". . . it seems that ice-shelf calving plays a role just as big". This again comes totally out of the blue and at the moment there is no way to check this statement as it is simply not described how calving is handled in the model. I also do not quite follow the explanation for this. Could the authors please elaborate on this?*

**Our response:** Calving in our model is done by a simple thickness threshold, where ice thinner than 50 m is calved out instantly. We have included this in our methods section (see our response to comment 1). Furthermore, in the rewriting of the discussion, this passage was removed. It no longer made sense to discuss calving there.

**39.** *L393 delete objective*
*L406-407 Delete last sentence (see comment above).*
**Our response:** We have removed the sentence and the word "objective".

**40.** *Comment hyphenation: I noticed that throughout the manuscript your use of hyphenation is inconsistent. You write ice-shelf calving, but then grounding line advance without hyphen. I am not sure what the TC policy is, but please make sure that you are at least consistent throughout the manuscript*
**Our response:** We thank the reviewer for noticing it, and have addressed the mistakes.

**Figures**

**41.** *Fig. 1: Please add a scale bar. Glacial index plots and their labels could be bigger.*
**Our response:** We have added a scale bar. The GI plots were removed from Fig 1 (see our response to comment 3).

**42.** *Fig. 2: Why do you show the time series until present-day? I think a zoom in into the period of interest would be better.*
**Our response:** Thank you for this suggestion. We have zoomed in to our period of interest, and added the GI plots from Fig. 1 in the original submission as Fig. 2b in this response letter.

**43.** *Fig. 3: It is really hard to see any differences in the upper panel (a-d) with the current colour scale. Also the grounding-line position should be made more prominent (thicker line or different color). In general there is too much white space and subplot labels (a-g) are too small. Please make each subplot bigger for better readability. Please also add a scale bar.*
**Our response:** The reviewer has a good point that it is hard to see differences in the upper panel, and a rescaling of the colorbar did not satisfactorily improve it. Thus, we have changed the figure to show only the control topography, and kept the difference plots to compare with the other geometries. We added thicker and colored lines for the grounding lines, which are plotted over their respective difference plot and over the control plot for an easier comparison. We have also added a scale bar as suggested. The same was applied to Fig. 10, which had the same style. Both figures were presented earlier in this letter as Figs. 9 and 10 (under our response to comment 35).

**44.** *Fig. 4: In the lower plot it looks like your model run for LR04 is not really in steady state or is your initial perturbation that large compared to your spin-up forcing? As mentioned above, I do not find the current y-axis units very intuitive for the lower panel. Labels (a,b) are too small.*
**Our response:** We never intended for it to be in steady state before this period, which is why we performed a thermal spin-up and gave it a relaxation period. Given the changes to the relaxation stage mentioned above, this figure has changed substantially, as shown in Fig. 3 of this response letter. A reference line indicating present-day Antarctic ice volume, suggested by Reviewer 2 (comment 36), helps put the presented numbers into perspective.

**45.** *Fig. 5: I do not really understand the point of this Figure as I do not get any information about the magnitudes of basal melting or the SMB. This Figure also needs a scale bar.*

**Our response:** We find that the changes incurred have improved the figure (see Fig. 7 in this letter, our responses to comments 26 and 29). We now show SMB for the grounded ice sheet, basal melting for the ice shelves, and added different hatching to where ablation occurs, and to where basal melting dominates over SMB at the ice shelves.

*46.* *Fig. 6: See Fig. 4*
*47.* *Fig. 7: Labels (a,b) are too small.*
*48.* *Fig. 8: Labels (a,b) are too small.*
**Our response:** We have increased the font size of all figures.

**References cited in this letter that were not listed under the original manuscript submission**

Barletta, Valentina R., et al. "Observed rapid bedrock uplift in Amundsen Sea Embayment promotes ice-sheet stability." *Science* 360.6395 (2018): 1335-1339.

Calov, Reinhard, and Ralf Greve. "A semi-analytical solution for the positive degree-day model with stochastic temperature variations." *Journal of Glaciology* 51.172 (2005): 173-175.

Holden, P. B., et al. "Interhemispheric coupling, the west Antarctic ice sheet and warm Antarctic interglacials." *Climate of the Past* 6.4 (2010): 431-443.

Maule, Cathrine Fox, et al. "Heat flux anomalies in Antarctica revealed by satellite magnetic data." *Science* 309.5733 (2005): 464-467.

Mitrovica, Jerry X., Natalya Gomez, and Peter U. Clark. "The sea-level fingerprint of West Antarctic collapse." *Science* 323.5915 (2009): 753-753.

Pollard, David, and R. M. DeConto. "A simple inverse method for the distribution of basal sliding coefficients under ice sheets, applied to Antarctica." *The Cryosphere* 6.5 (2012b): 953.

Waelbroeck, Claire, et al. "Sea-level and deep water temperature changes derived from benthic foraminifera isotopic records." *Quaternary Science Reviews* 21.1-3 (2002): 295-305.

Yang, Haijun, and Jiang Zhu. "Equilibrium thermal response timescale of global oceans." *Geophysical research letters* 38.14 (2011).

---

## Author Comment (AC2) · 31 Aug 2020

**Dear editor, dear reviewer,**

We thank the reviewer for their constructive, insightful and helpful evaluation which we feel helped to improve the manuscript. This instigated additional modeling that resulted in numerous refinements, and significant upgrades to the model description and discussion sections. Below, we provide a point-by-point response to each comment, which we numbered in red for easier reference. Our response is structured as follows: Referee comment (*in black italics*), author's response (in green), and proposed changes in the original manuscript text (*in blue italics*) where significant rewriting was done to include the suggested changes. We also add to the end of each figure caption (in blue) their proposed numbering after the changes made to the original submission of the manuscript.

Finally, we would like to draw the reviewer's attention to the correct reference to the first author's last name, as it is "Mas e Braga" and not "Braga".

**General Comments**

**1.** "I find inconclusive the set of experiments to determine whether or not the duration of the interglacial is responsible for AIS retreat rather than a warm peak as for MIS5. This is because the index derived from ice core records mainly impact on the oceanic forcing of the simulation which generates a tipping point. Once the tipping point is crossed, then the duration of the interglacial does not matter at all to explain the amplitude of ice sheet retreat in the simulations. In simulations using Vostok-GI, ice volume is lower but because the GI does not yield too warm temperature."

**Our response:** The reviewer has a very good point, which we missed in our original submission. We have made sufficient changes to our analyses to address the possible tipping point mentioned by the reviewer. First, we have added a figure (Fig. 1a,b in this letter) that shows the oceanic forcing under the main ice shelves as requested in comments 53 and 54, compared to the Summer atmospheric temperatures. Based on what this figure shows, there is indeed a tipping point where the ocean starts to rapidly warm up at around 412 ka, reaching temperatures up to 2 °C warmer than the ice shelf base, as expressed by the thermal forcing (i.e., the difference between ocean temperatures and ice shelf temperatures at the ice/ocean interface). To investigate whether this is a tipping point caused by the increase in Tforc, we performed four new experiments. Two are based on the EDC ice core, one where we keep the climate constant before and after the suspected tipping point (at 416 and 410 ka respectively). The other two are based on the Vostok ice core, one where we keep the climate constant at its peak GI value, and one where we rapidly move the climate from its peak back to constant prepeak conditions (although still warmer than the 412 ka conditions). These are shown below in Fig. 2. We will reformulate our discussion in light of these new results, which essentially show that the duration of warming was key for instigating strong WAIS retreat, while warming intensity (peak) allowed the retreat to be accelerated or delayed. There is indeed a threshold of 1.5 °C which must be crossed for the WAIS to collapse.

Figure 1: Evolution throughout MIS11 for each CFEN member averaged over specified Antarctic ice shelves for (a) Summer surface air temperature [°C]; (b) thermal forcing under the ice shelves (i.e., ocean temperature minus ice base temperature, in °C); and (c) sea level contribution by EAIS and WAIS. Colours denote the respective CFEN member, while line styles denote each ice shelf (panels a,b), and each ice sheet (panel c). DML refers to an average for ice shelves along the Dronning Maud Land between 27°W and 30°E. This is Fig. 9 after the revisions to the manuscript.

---

## Author Response (AR1)

Dear Prof. Whitehouse,

Please find enclosed a thoroughly revised manuscript based on the excellent reviews we were provided with. We also provide the point-by-point response to each reviewer and a "latexdiff" version highlighting changes from the original submission. You will notice that some minor changes occurred relative to those in our original response to reviewers. This is due to a final extensive internal revision to ensure a more fluid reading and a better connected text. We hope that you can appreciate this difference.

The study now includes an assessment of the effect of horizontal resolution (Fig. S15), which we proposed to include in response to your initial evaluation before the manuscript was accepted for reviews. We performed simulations at several resolutions, refining from 20 km (which we present in the main text) up to 10 km. Simulations at 16 and 15 km are already finished and included in the mentioned figure. Due to technical issues that have been already solved, the computationally expensive simulations at 10 and 12 km had to be re-run and are not yet complete. However, we can already see that none of these additional simulations shows significant changes that affect the conclusions of our study.

We hope that you find our response satisfactory, and the updated manuscript further strengthened.

Yours Sincerely,

Martim Mas e Braga (corresponding author) Dear editor, dear Dr. Schanwell,

We thank the reviewer for his constructive, insightful and helpful evaluation which we feel helped to improve the manuscript. This instigated additional modeling that resulted in numerous refinements, and significant upgrades to the model description and discussion sections. Below, we provide a point-by-point response to each comment, which we numbered in red for easier reference. Our response is structured as follows: Referee comment (*in black italics*), author's response (in green), and proposed changes in the original manuscript text (*in blue italics*) where significant rewriting was done to include the suggested changes. We also add to the end of each figure caption (in blue) their proposed numbering in the revised version of the manuscript.

**Main concerns**

**1.** "[...] I suggest to expand section 2.1 to add this required information. To be more specific, what type of stress balance does SICOPOLIS use? What kind of basal friction law do you apply? I know you list the parameters in Table 1, but without the corresponding equation, they are rather useless. Does your basal friction coefficient vary spatially and/or temporally? What are your boundary conditions for your enthalpy equation (e.g. do you specifiy a geothermal heat flux? Is it spatially constant?)? How do you treat calving in the model? There are a number of ways how to parameterise this. Since you talk about this in your results, it is essential to know how this is handled in your model.

Also you should mention with what geometry your initialise your model. I believe it is with present-day geometry, but with which dataset (Bedmap2, Bedmachine)? Do you take the bedrock and ocean floor topography from the same dataset?"

**Our response:** We have expanded the model description, including the requested additional information, as can be seen below:

"For our experiments we employ the 3D thermomechanical polythermal ice-sheet model SICOPOLIS (Greve, 1997, Sato & Greve, 2012) with a 20 km horizontal grid resolution and 81 terrain-following layers. It uses the one-layer enthalpy scheme of Greve & Blatter (2016), which is able to correctly track the position of the cold-temperate transition in the thermal structure of a polythermal ice body.

The model combines the Shallow Ice Approximation (SIA) and Shelfy Stream Approximation (SStA) using (c.f. Bernales et al., 2017a, Eq. 1)

 $U = (1 - w) \cdot u_{sia} + u_{ssta}$

where U is the resulting hybrid velocity,  $u_{sia}$  and  $u_{ssta}$  are the SIA and SStA horizontal velocities, respectively, and w is a weight computed as

$$w(|u_{ssta}|) = \frac{2}{\pi} \arctan\left(\frac{|u_{ssta}|^2}{u_{ref}^2}\right)$$

where the reference velocity,  $u_{ref}$ , is set to 30 ma-1, marking the transition between slow and fast ice. This hybrid scheme reduces the contribution from SIA velocities mostly in coastal areas of fast ice flow and heterogeneous topography, where this approximation becomes invalid. Basal sliding is implemented within the computation of SStA velocities as a Weertmantype law (cf. Bernales et al., 2017a, Eqs. 2--6). The amount of sliding is controlled by a fixed, spatially varying map of friction coefficients that was iteratively adjusted during an initial present-day equilibrium run (cf. Pollard & DeConto, 2012b), such that the grounded ice thickness matches the present-day observations from Bedmap2 (Fretwell et al., 2013) as close as possible. Sliding coefficients in sub-ice shelf and ocean areas are set to  $10^5 \text{ ma}^{-1}\text{Pa}^{-1}$ , representing soft, deformable sediment, in case the grounded ice advances over this region. The initial bedrock, ice base, and ocean floor elevations are also taken from Bedmap2. Enhancement factors for both grounded and floating ice are set to 1, based on sensitivity tests in Bernales et al. (2017b). This choice provides the best match between observed and modelled ice thickness for this hybrid scheme, similar to the findings in Pollard & DeConto (2012a).

Surface mass balance is calculated as the difference between accumulation and surface melting. The latter is computed using a semi-analytical solution of the positive degree day (PDD) model following Calov & Greve (2005). Near-surface air temperatures entering the PDD scheme are adjusted through a lapse rate correction of 8.0  $^{\circ}C$  km-1 to account for differences between the modelled ice sheet topoaraphy and that used in the climate model from which the air temperatures are taken. For the basal mass balance of ice shelves, we use a calibration scheme of basal melting rates developed in Bernales et al. (2017b) to optimise a parameterisation based on Beckman & Goosse (2003) and Martin et al. (2011) that assumes a quadratic dependence on ocean thermal forcing (Holland et al., 2008; Pollard & DeConto, 2012; Favier et al., 2019). This optimised parameterisation is able to respond to variations in the applied Glacial Index (GI, Sect. 2.2) forcing. A more detailed description of this parameterisation is given in Sect. 1 of the supplementary material. In our experiments, we prescribe a time lag of 300 years for the ocean response to GI variations, which is considered the most likely lag in response time of the ocean compared to the atmosphere in the Southern Ocean (Yang & Zhu, 2011). At the grounding line, the basal mass balance of partially floating grid cells is computed as the average melting of the surrounding, fully floating cells, multiplied by a factor between 0 and 1 that depends on the fraction of the cell that is floating. This fraction is computed using an estimate of the sub-grid grounding line position based on an interpolation of the current, modelled bedrock and ice-shelf basal topographies. At the ice shelf fronts, calving events are parameterised through a simple thickness threshold, where ice thinner than 50 m is instantly calved away.

Glacial isostatic adjustment is implemented using a simple elastic lithosphere, relaxing asthenosphere (ELRA) model, with a time lag of 1 kyr and flexural rigidity of  $2.0 \times 10^{25}$  Nm, which Konrad et al. (2014) found to best reproduce the results of a fully-coupled ice sheet–self-gravitating viscoelastic solid Earth model. The geothermal heat flux applied at the base of the lithosphere is taken from Maule et al. (2005) and is kept constant. All relevant parameters used in the modelling experiments are listed in Table 1."

"2. Could you please motivate the ensembles or parameters changes that you are investigating a bit more? As it stands now, it seems like you picked a number of parameters, but there also could an argument be made for a bunch of other parameters to be varied."

**Our response:** We picked these ensembles as they are inherent sources of uncertainty that were not addressed by any previous studies that included MIS11. We also performed additional tests to support our parameter choices (such as ocean temperature, lag in its response, and the choice of climate model), which we included in the supplementary material. We justify the choice of ensembles in the last paragraph of the introduction:

"For this purpose, we perform five ensembles of numerical simulations of the AIS evolution and focus on aspects that remain unaddressed by previous studies. We evaluate the impact on resulting ice volume and extent of the choice of proxy records (including their differences in signal intensity and structure), the choice of sea level reconstruction, and of uncertainties in assumptions regarding the geometry of the AIS at the start of MIS11c."

**"3.** I find most of the figures (e.g. 4, 6,7,9) not very informative. Looking at integrated quantities is OK, but having five Figures like that is too much. I suggest to combine them into a Figure with several panels. I also find it hard to judge in these volume plots whether differences are small or large (Is 2000 km3 a lot?). Maybe better to plot it in percent normalised to your starting volume? Also just because your ice volume is similar does not mean you cannot have regional differences in grounding-line position or ice thickness. For example on P16L279 you state "... show similar retreat rates..." but I cannot find a Figure where this is actually shown. So I suggest to add some Figures, where we can also look at some spatial differences (a few suggestion in the technical corrections below). For example, you could plot some grounding-line positions from different simulations in 2D on top of each other to see the differences in retreat or lack thereof. I also encourage the authors to discuss their results more in depth. For example, they state in L276ff that different initial ice sheet configurations converge to the same geometry for the same climate forcings. This alone is quite surprising to me and

at least warrants a discussion why potential feedback mechanisms (e.g. stabilising grounding-line on topographic height) are not triggered in these simulations?"

**Our response:** We have added a compilation of all different GIs as a panel in Fig. 2, which allows us to remove them from Fig. 1 (see Figs. 1 and 2 below). Regarding figures 4, 6, 7, and 9, they were restructured, with their (b) panels being merged into a single figure, and added a line that represents present AIS volume, as suggested by Reviewer 2 (see Fig. 3 below). We have added a new figure (see Fig. 4 below) where we show the grounding lines for each of our core experiments at times of interest: 420, 415, 410, and 405 ka. Regarding the fact that different configurations converge to the same geometry, we have found a tipping point at 412 ka (as pointed out by Reviewer 2 in his comment 47), where the ocean forcing under the main ice shelves (cf. Fig. 8) is strong enough to drive ice sheet retreat in all geometry scenarios. There are two grounding-line stabilising feedbacks not included in our current version of the model: (*i*) a local sea-level drop caused by a reduced gravitational attraction of a shrinking ice sheet (e.g. Mitrovica et al., 2009), and (*ii*) the observed faster rebound of the crust due to a lower mantle viscosity in some WAIS locations (Barletta et al., 2018). Even though our ELRA model is set up with a relatively fast response time of 1 kyr (compared to the standard 3 kyr), the resulting bedrock uplift is still not able to trigger a stabilizing effect that compensates for the strong ocean-driven retreat. These feedback mechanisms during MIS11c could be further investigated through the utilization of an Earth-ice coupled model, which is certainly an interesting topic for future research. These points are incorporated in the end of our Discussion section:

There are, however, two stabilising feedbacks which are not incorporated in our model: (i) a local sea-level drop caused by a reduced gravitational attraction of a shrinking ice sheet (e.g., Mitrovica et al., 2009), and (ii) the observed faster rebound of the crust due to a lower mantle viscosity in some WAIS locations (Barletta et al., 2018). The first effect is probably small based on our model's insensitivity to sea-level changes over these time scales, but we have been unable to robustly test the effect of a faster rebound on AIS response during MIS11c. However, we note that our ELRA model is set up with a relatively short response time of 1 kyr, for which the resulting bedrock uplift is still not able to trigger a stabilizing effect large enough to prevent WAIS collapse.

Figure 1: Surface topography of the AIS at the start of our core experiments (425 ka), based on a calibration against Bedmap2 (Fretwell et al., 2013; see Sect. 2.1). The locations mentioned in the text, including the drilling sites of the ice (circles) and sediment (red diamonds) cores on and around Antarctica, are showcased. This is Fig. 1 after the revisions to the manuscript.

---

## Referee Report (RR1)

**Review of revised manuscript by Mas e Braga et al. "Sensitivity of the Antarctic ice sheets to the peak warming of Marine Isotope Stage 11"**

November 6, 2020

**General comments:**

This is the second review of Mas e Braga et al. following an initial round of reviews. I think the manuscript is much improved in comparison to the previous round of reviews and most of the points raised by the reviewers in the first review have been addressed. However, there still remain a few questions/suggestions for improvements listed below that I would like to see addressed before publication.

**Scientific comments:**

L79-82 In the first review I asked to motivate the choice of ensembles and parameter changes a bit better and the authors have added a paragraph that discusses this. However, I feel the argument that these parameters have not been addressed before a bit weak. Many other parameters (e.g. type of basal sliding law, different ocean parameterisation) could also have been chosen, so it would be good to state why you selected these particular parameters. Are they what you consider most important for sea-level rise or are they the most unconstrained? Please clarify this.

L181 Could you briefly say why you picked the EDC record for your ensemble here?

Sections 3.3. and 3.4 I think you could combine these two sections and simply state that they are not important for ice-sheet volume differences. Especially regarding section 3.3., volume differences are basically non-existent. At least I cannot see much difference at all in Fig. 4c. Similar arguments apply for Fig. 4e. Also, regarding the difference in floating ice volume at the beginning of the simulation, is that a result of forcing the same initial geometry with different sea-levels or why do they not start from the same value?

Caption of Fig. 8: I think that your statement: "Everywhere where $Q_{bm}$ >SMB, ice shelves are thinning" is questionable. I think if you think about a Lagrangian framework this is correct, but if you think about it in an Eulerian framework (more common in ice-sheet modelling), you could have higher basal melt rates than SMB, but still have local thickening because thicker is is being advected from upstream. This is also true if you consider entire ice shelves. You can have a negative budget from SMB and basal melting, but still gain mass, because thicker ice is being advected from upstream. So I recommend deleting this statement.

Paragraph starting at L304: I think it is not surprising that ocean melting does not do as much to the EAIS as it does to the WAIS. WAIS is a marine ice sheet with large shelves providing a lot of buttressing, while the EAIS has only small ice shelves which provide less buttressing and is predominately not marine-based.

Paragraph starting L340: I think here it would be good to add a qualifier that these numbers are for your particular ocean melt parameterisation. If you use a different paramerisation (e.g. linear relation), these thresholds would most certainly change as well.

L349-354 I do not find your different resolution experiments particularly helpful or well thought-through. So you test, 20, 16, and 15 km. This seems like a random choice of model resolutions. Especially the step from 16 to 15 km, is really small. If you want to do this more rigorously, you would have to do a convergence study (see for example Cornford et al. 2016 or Schannwell et al. 2018). I am not suggesting that you should do this, but I think you are definitely overstating your results and should be more cautious with your conclusions. In your Fig S15c, it is hard to tell because it is rather small, but you definitely see differences in sea level, even from 16 to 15 km. So your results are certainly to a degree mesh resolution dependent. From my own experience, if you increase your resolution to 10 km you start resolving ice streams a lot better and you would probably see more differences. I think that your different mesh resolutions are not fine enough to make the claim that results are mesh resolution independent. Rather, from the evidence that I see the contrary is the case. If you cannot run higher resolved simulations with your model because of computational restrictions that is fine, but it should be clearly stated.

In you conclusions you state "...WAIS collapse was caused by the duration rather than the intensity of warming...", but in the discussion you say that both conditions have to be met and that even a shorter, but more intense ocean warming may also lead to WAIS collapse.

**Figure comments:**

The main point that needs improving are Figure sizes, Figs 3, 5, 7, 8 are plotting continent wide grounding lines. But each panel is so small, it is nearly impossible to see any differences. So please make each panel a lot bigger. If it helps, you could change to a 2x2 panel format. You can also use the full width of the page to make them bigger. For example, Figure sizes are much better in the supplement.

Figure 8:
I am sorry, but the hatching where basal melting is dominating SMB and vice versa is not visible. Even with a 300% zoom it is hard to see. You would probably have to have a zoom-in into the regions you talk about in the text in an additional Figure to see this.

Please also make Figure 10 a lot bigger. The insets are so small I had to use 300% zoom to see everything. No chance on the printout.

**Technical corrections:**

L98 "controlled by a temporally fixed"

L278 "CFEN equivalent run" maybe

L304 "close to grounding lines"

**References**

Cornford, S., Martin, D., Lee, V., Payne, A., and Ng, E. (2016). Adaptive mesh refinement versus subgrid friction interpolation in simulations of Antarctic ice dynamics. Annals of Glaciology, 57(73), 1-9. doi:10.1017/aog.2016.13

Schannwell, C., Cornford, S., Pollard, D., and Barrand, N. E.: Dynamic response of Antarctic Peninsula Ice Sheet to potential collapse of Larsen C and George VI ice shelves, The Cryosphere, 12, 2307–2326, https://doi.org/10.5194/tc-12-2307-2018, 2018.

Sincerely, Clemens Schannwell

---

## Editor Decision (ED1)

Dear authors,

Thank you for submitting a revised version of your manuscript and for addressing all the points raised during the previous round of reviews. You have responded to all the reviewers' technical queries, the readability of the figures has been improved, and additional results demonstrating that key findings do not depend on model resolution are included in the supplementary material (although note that the caption to figure S15 needs updating).

I have read the revised version of the manuscript and a small number of points require clarification:

1. **MIS11 and MIS11c**: these terms appear to be used somewhat interchangeably, please clarify whether there is a difference between them, and if there is, then check that terms are used consistently throughout the text.

2. **Timing of the highstand**: it seems to be assumed that the MIS11c sea level highstand was at 405 ka. If this is the case, please state this clearly somewhere, referring to supporting evidence. If the highstand is not independently constrained to be at 405 ka, then do your results perhaps suggest that the lower bound for the AIS contribution to the highstand is 4 m – based on the Vostok scenario (fig. 6), where the sea level contribution is ~4 m at 410 ka?

3. **CCSM3 cold bias**: the statement on lines 304-305 requires further clarification. If I have understood the reviewer correctly, the CCSM3 LGM climate is too cold (even after correcting for the lapse rate), but the CCSM3 PI climate is relatively accurate. Due to the approach used to create the GI, this therefore leads to a cold bias for positive GI values and a warm bias for negative GI values (i.e. during an interglacial) – as shown in figure S14. Is this correct?

Additional minor suggested corrections are listed below. Once these, and the issues raised above, are addressed, I would be happy to accept this article for publication in The Cryosphere.

Pippa Whitehouse
Associate Editor

Minor points

Line 8: 'contributed 3.2-8.2 m to…'

Line 10: delete 'further'

Lines 12-13: it is not clear how the climate signal is linked to global sea level, or what it means to 'match the recorded global sea level highstand'. Text could be tightened and perhaps linked more closely to information in the final sentence

Line 18: delete 'ka'

Line 51: make it clear that the Dry Valley moraines are interpreted to indicate local ice advance

Line 68: 'apeak' – space missing

Line 79: To improve the structure of this paragraph, I suggest first stating that model results depend on forcings, boundary conditions, model parameters etc. You can then summarise which areas have previously been studied, before highlighting which aspects you will focus on.

Line 84: 'could help guiding' – check grammar

Line 85: Awkward sentence, suggest, "We evaluate the impact of the following on AIS volume and extent during MIS11c: the choice of…"

Figure 1: I do not see any red diamonds indicating sediment cores

Line 96: this looks like eq. 9 in Bernales et al. (2017a), not eq. 1. Should $u_{ssta}$ be multiplied by *w*?

Line 127: your approach does not include all aspects of glacial isostatic adjustment (an important component is the spatial variation in sea surface height), suggest replacing with 'bed deformation'

Line 142: 'When analysing the results, we ignore…'

Line 144: Clarification needed because the EDC record was not used to force *all* the ensemble runs, e.g., it was not used to force all the CFEN experiments

Line 157: '…assess the impact of similarities and differences…'

Table 2 caption: clarify that 'Age (ka)' relates to *LGM* reference values

Line 187: delete 'Mean'. Also, given that your approach does not account for local gravitational perturbations to sea surface height, I suggest adding a sentence: "We approximate the sea level forcing applied at the boundaries of the ice sheet using global mean sea level reconstructions."

Lines 214-221: this text describes the initial ice sheet configurations (gmt1-gmt3) for the EDC case (shown in figure 3). Does it also hold for the cases when DF and Vostok forcing are used?

Lines 218 and 219: '…than the control…'

Table 4: reference to $\delta X_{Hol}$ is perhaps left over from an earlier version of the manuscript?

Line 263: you state above (line 258) that using the LR04 average values gives a 3.4% smaller ice sheet at 402 ka, and here you state that using the EDC average value gives a 2.3% larger ice sheet at 402 ka. However, in figure 4b, the orange solid/dashed lines are much closer to each other at 402 ka than the black solid/dashed lines - please check calculations

Line 271: 'It directly reflects their effect' – references to 'it' and 'their' are ambiguous

Line 284: 'different initial geometries'

Line 285: 'The latter two…' – check, I think it is Totten and Dibble that are thicker, with Cook thinner

Line 318: 'the former two' – not clear what this refers to

Line 323: 'the different ice-sheet configurations' – make it clear that you are talking about model runs forced by the same ice core record, but with different initial ice sheet configurations

Line 340: clarify that the values relate to ocean temperatures

Line 356: 'WAIS collapse was triggered' – more caution needed in the language used, it is not proven that WAIS collapsed during the LIG

Line 370: 'ested' -> 'tested'

Figure 10 caption: (b,e) -> (b,f)

Line 391: 'when comparing their results' – check the logic in this sentence

Lines 397 and 399: suggest 'interval' -> 'range'

Line 398: 'ice core experiments'

line 444: the tense of the final sentence is odd, suggest "We found that this threshold needed to be sustained for at least 4 kyr for strong WAIS ice retreat to be triggered."

---

## Author Response (AR2)

Dear Editor,

We would like to thank the reviewers and yourself for the excellent and constructive feedback on our revised manuscript. Below we provide a point-by-point response to all comments. The original comments are kept in black and are numbered (in red). We provide our responses in green, and show manuscript revisions in *blue italics*. We also provide a "latexdiff" version of the manuscript, where the changes listed in the responses are highlighted.

We would like to emphasise that we have enlarged all the figures mentioned by the reviewers, and changed single-row plots to two-row plots where suggested. Reviewer 1 further notes in his comment 11 that Figure 10 demanded excessive digital enlargement for reading, and mentions in comment 9 that we could use the full width of the page to make the figures larger. However, in the LaTeX template two-column figures are set to a width of 12 cm, although using 18 cm still seems to keep the figure within the same width as the text. We therefore submit an 18 cm-wide Figure 10, but would be happy to submit it in another size if this is not permitted.

We hope that you find our responses satisfactory, and that you will accept our revised version for publication.

Best regards,

Martim Mas e Braga (corresponding author)

**Editor's comments**

**1.** Lines 4-6: check the logic of this sentence – statements are not really linked
We revised the sentence slightly, which now reads (L4):
*"Ice core data indicate that warmer-than-present temperatures lasted for longer than during other interglacials. However, the response of the Antarctic ice sheets and their contribution to sea level rise remain unclear."*

**2.** Line 10-11: confusing use of 'sea level reconstructions' in two different senses
We slightly rephrased the sentence for clarity. It now reads (L9):
*"In the case of a West Antarctic Ice Sheet collapse, which is the most probable scenario according to far-field sea level reconstructions, the range is further reduced to 6.7-8.2 m independently of the choices of external sea-level forcing and millennial-scale climate variability."*

**3.** Line 283: I think the opening statement is based on experiments driven by different proxy records, rather than the interpretation of the proxy records themselves
The statement is indeed unclear, thank you for pointing that out. We clarified it by writing (L293):
*"We base this statement on results from experiments forced by different proxy records with significant differences in their structure during the MIS11c peak warming."*

**4.** Line 322: when describing experiment (ii) I suggest emphasising that 410 ka lies after the sudden increase in ocean temperature but before the maximum is reached
Thank you for the suggestion. We added this information in a parenthesis (L334):

*"(i.e., just after the sudden increase in ocean temperatures, but before the maximum is reached; cf. Fig 9b)"*

**5.** General: use of 'this' is occasionally ambiguous, e.g. lines 38, 191, 340
We list below the changes made to the mentioned lines. In L340, we note that changes were also made as a response to comment 8 of Reviewer 1.
**L37:** *"The unusual length of MIS11c and a transition […]"*
**L201:** *"Although the similarity to the modern AIS configuration has been loosely inferred from sedimentary (Capron et al., 2019) and ice-core (EPICA Community Members, 2004) proxy records, to our knowledge there is no direct evidence to support this claim […]"*
**L353:** *"The critical warming of 0.4 ℃ we found for MIS11c is close to the equilibrium model results […]"*

**6.** General: please clarify how you convert ice volume to sea level contribution
We added a clarification of our sea level contribution conversion in the methods (L132):

*"Sea-level contribution at a given time step is computed in SICOPOLIS as the difference in total ice volume above flotation between the ice sheet at the time step and the spun-up Pre-Industrial ice sheet. When computing ice volume, differences in bedrock elevation between the two ice sheets are accounted for by using a common reference bedrock elevation in all time steps. We further correct for the projection effect on the horizontal grid area."*

**7.** General: please clarify how you define WAIS collapse
We consider WAIS to have collapsed when the Weddell, Ross, and Amundsen seas become interconnected. We clarify this after the first mention of WAIS collapse in L245:
*"However, it recovers more quickly than the EDC and DF experiments as soon as the peak warming is over and the climate starts to shift back to PI conditions, without a WAIS collapse (we consider the WAIS to have collapsed when the Weddell, Ross, and Amundsen seas become interconnected; Fig. 5)."*

**8.** Supp. Fig. S2: please clarify whether the forcings shown relate to glacial or interglacial conditions
We added to this information to the caption of Fig. S2:
*"Comparison of Pre-Industrial CCSM3 forcings (right) to reference data (left) from RACMO2"*

**9.** Supp. Fig. S13: please clarify what is meant by 'anomalies'
We have added an explanation of anomalies to the caption of Fig. S13, which now reads:
*"(a-f) Forcing fields used to construct the climate forcings used in this study. Left fields show PI mean states, while right fields show applied anomalies (i.e., LGM−PI for temperatures, and LGM÷PI for precipitation). Anomalies are multiplied by the GI and then combined with the left-side fields following Eqs. (4) and (5)."*

**Reviewer 1's comments**

**Scientific comments**

**1.** L79-82 In the first review I asked to motivate the choice of ensembles and parameter changes a bit better and the authors have added a paragraph that discusses this. However, I feel the argument that these parameters have not been addressed before a bit weak. Many other parameters (e.g. type of basal sliding law, different ocean parameterisation) could also have been chosen, so it would be good to state why you selected these particular parameters. Are they what you consider most important for sea-level rise or are they the most unconstrained? Please clarify this.

We decided to focus on aspects that were external to the model, (i.e., that mainly impact boundary and starting conditions) rather than aspects that are more model dependent. For this same reason, we refrained from calling them "parameters". We chose those aspects expecting that our findings regarding the different choices considered would be of use to a wider community. We add this justification, refining the last paragraph of the introduction (L79):

*"The sensitivity of ice volume changes across glacial-interglacial time scales to model parameters was extensively explored by Albrecht et al. (2020). DeConto & Pollard (2016) carried out a large ensemble analysis for the LIG and the Pliocene, where parameters related to ice-shelf loss were constrained according to their ability to simulate target ranges of sea-level contribution. Simpler flow-line models have also been used to evaluate uncertainties in basal conditions (Gladstone et al., 2017) and flow-law parameters (Zeitz et al., 2020). Here, we perform five ensembles of experiments that focus on choices that are external to the numerical model, and could help guiding other modelling efforts on the choice of forcings and boundary conditions. We evaluate the impact on AIS volume and extent during MIS11c of the choice of proxy record (including their differences in signal intensity and structure), the choice of sea level reconstruction, and of uncertainties in assumptions regarding the geometry of the AIS at the start of MIS11c."*

In the discussion, we further acknowledge the fact that we do not test for parameters such as the choice of sliding law (see response to comment 7) or different ocean parameterisations, and that they could impact our results (also mentioned in comment 6, and jointly answered in comment 7).

**2.** L181 Could you briefly say why you picked the EDC record for your ensemble here?
We use the EDC record because our thermal spin-up was also performed using data from this record. We justify our choice for the EDC as the reference ice core towards the end of Sect. 2.1 (L144):

*"The EDC ice core was chosen for the thermal spinup and as forcing for the ensemble runs, because it spans the longest period among the three ice cores tested, while still providing a relatively high temporal resolution."*

**3.** Sections 3.3. and 3.4 I think you could combine these two sections and simply state that they are not important for ice-sheet volume differences. Especially regarding section 3.3., volume differences are basically non-existent. At least I cannot see much difference at all in Fig. 4c. Similar arguments apply for Fig. 4e. Also, regarding the difference in floating ice volume at the beginning of the simulation, is that a result of forcing the same initial geometry with different sea-levels or why do they not start from the same value?

We agree that both sections were very short, and have merged them. Yes, the differences in floating ice volume at 425 ka are a result from forcing the same initial geometry with different sea-level reconstructions. As explained in the methods (paragraph starting at L138), they do not start from the same point in Figs. 4d and e because the starting point, under a common geometry, is at 425 ka (see Fig. S12 for an example for CFEN).

**4.** Caption of Fig. 8: I think that your statement: "Everywhere where Q bm >SMB, ice shelves are thinning" is questionable. I think if you think about a Lagrangian framework this is correct, but if you think about it in an Eulerian framework (more common in ice-sheet modelling), you could have higher basal melt rates than SMB, but still have local thickening because thicker is is being advected from upstream. This is also true if you consider entire ice shelves. You can have a negative budget from SMB and basal melting, but still gain mass, because thicker ice is being advected from upstream. So I recommend deleting this statement.

That is a very good point. We wanted to better guide the reader in terms of the magnitude of each process (because of the different sign conventions for each), but introduced a dubious statement. We have therefore removed it.

**5.** Paragraph starting at L304: I think it is not surprising that ocean melting does not do as much to the EAIS as it does to the WAIS. WAIS is a marine ice sheet with large shelves providing a lot of buttressing, while the EAIS has only small ice shelves which provide less buttressing and is predominately not marine-based.

We agree with this statement, and never intended to claim otherwise. We include the provided remarks and rephrased the end of the paragraph (L322):

*"This observed tipping point at 412 ka also explains why the different ice-sheet configurations all follow the same trend from that moment onwards (Fig. 6), and why the evolution of WAIS and EAIS sea level contributions diverge. As ocean forcing becomes the main driver of ice-sheet retreat, it has a much larger impact on marine-based portions of the ice sheet. Around most of the EAIS (except for the Amery Ice Shelf), ice shelves are small and provide little buttressing. Hence, because most of the EAIS is grounded above sea level, its sub-shelf melting is not high enough to force grounding line retreat as strongly as in the WAIS. As a consequence, ice melt is dominated by surface ablation at the ice-sheet fringes (cf. hatched patterns in Fig. 8)."*

**6.** Paragraph starting L340: I think here it would be good to add a qualifier that these numbers are for your particular ocean melt parameterisation. If you use a different paramerisation (e.g. linear relation), these thresholds would most certainly change as well.

We inserted the suggested qualifier while reformulating this paragraph as a response to comment 7.

**7.** L349-354 I do not find your different resolution experiments particularly helpful or well thought-through. So you test, 20, 16, and 15 km. This seems like a random choice of model resolutions. Especially the step from 16 to 15 km, is really small. If you want to do this more rigorously, you would have to do a convergence study (see for example Cornford et al. 2016 or Schannwell et al. 2018). I am not suggesting that you should do this, but I think you are definitely overstating your results and should be more cautious with your conclusions. In your Fig S15c, it is hard to tell because it is rather small, but you definitely see differences in sea level, even from 16 to 15 km. So your results are certainly to a degree mesh resolution dependent. From my own experience, if you increase your resolution to 10 km you start resolving ice streams a lot better and you would probably see more differences. I think that your different mesh resolutions are not fine enough to make the claim that results are mesh resolution independent. Rather, from the evidence that I see the contrary is the case. If you cannot run higher resolved simulations with your model because of computational restrictions that is fine, but it should be clearly stated.

In our reply to the first round of revisions, we noted that experiments at higher resolutions (12 and 10 km) were still underway and would be finished in time to be included in the following round of reviews. We have now included them to the supplement. Nevertheless, the reviewer brings a fair point regarding our statement about the sensitivity of our results to grid resolution. Based on the results of all higher resolution runs, and on comments 1 and 6, we rephrased with the following (L366):

*"Moreover, AIS minimum extent and the timing of WAIS collapse are robust regardless of model resolution (Fig. S15). A set of simulations performed with several resolutions (from 20 to 10 km) showed virtually the same changes in ice-sheet extent, and modest variations in ice volume, which amount to a spread of 1.2 m s.l.e. in sea level contribution at 405 ka. Alternative sliding laws or sub-shelf melting parameterisations, for example using a linear dependence of sub-shelf melt to ocean thermal forcing, or applying a more physically realistic approach (e.g., Reese et al., 2018) were not ested, and could influence our results. For example, numerical modelling studies in which the WAIS did not collapse during MIS11 were acknowledged to be less sensitive to the ability of ocean temperatures to drive basal melting (e.g., Pollard & DeConto, 2009; Tigchelaar et al., 2019). Finally […]"*

**8.** In you conclusions you state "...WAIS collapse was caused by the duration rather than the intensity of warming...", but in the discussion you say that both conditions have to be met and that even a shorter, but more intense ocean warming may also lead to WAIS collapse.

We understand how it might be confusing when looking at these two statements isolated. We state that both criteria need to be met, and when comparing to other studies we remark that the duration is specific to MIS11c. We then acknowledge that a stronger warming is also able to cause a WAIS collapse – which was most likely the case for LIG. We tried to make this distinction clearer by rewriting the paragraph starting at L353, also taking into account the changes presented above as a response to comment 7:

*"The inferred critical warming of intermediate-depth ocean temperatures of 0.4 ºC for MIS11c is close to the equilibrium model results in Garbe et al. (2020), but lower than results from Turney et al. (2020) for the AIS retreat during the LIG. While the former study shows a strong WAIS retreat is already possible for an ocean warming of 0.7 ºC, the latter identifies a tipping point at 2 ºC warming in ocean temperatures. In other interglacials, such as the LIG, WAIS collapse was triggered by ocean warming with a higher intensity and of shorter duration than during MIS11c (Dutton et al., 2015, Turney et al., 2020), since a stronger rate of warming can drive ice retreat at a much faster pace. Thus, WAIS collapse during MIS11c was likely attained because ocean temperatures exceeded a modest threshold for long enough (over 4 kyr)."*

**Figure comments:**

**9.** The main point that needs improving are Figure sizes, Figs 3, 5, 7, 8 are plotting continent wide grounding lines. But each panel is so small, it is nearly impossible to see any differences. So please make each panel a lot bigger. If it helps, you could change to a 2x2 panel format. You can also use the full width of the page to make them bigger. For example, Figure sizes are much better in the supplement.

We changed all 4x1 figures to a 2x2 format, and increased font size from 16 to 18 pt. Regarding using the full width of the page, please note our comment at the beginning of this letter. If permitted, we would happily increase the other figure widths to full page as well.

**10.** Figure 8: I am sorry, but the hatching where basal melting is dominating SMB and vice versa is not visible. Even with a 300% zoom it is hard to see. You would probably have to have a zoom-in into the regions you talk about in the text in an additional Figure to see this.

We changed the figure to a 2x2 panel and increased font size to 18 pt. We note that the different patterns of hatching were perfectly visible at 126% zoom on a 22-inch monitor.

**11.** Please also make Figure 10 a lot bigger. The insets are so small I had to use 300% zoom to see everything. No chance on the printout.
We have enlarged the figure, as noted in the beginning of our response letter.

**Technical corrections**
**12.** L98 "controlled by a temporally fixed"
**13.** L278 "CFEN equivalent run" maybe
**14.** L304 "close to grounding lines"
We have used all suggestions above.

**Reviewer 2's comments**

**1.** Line 10: add a word to "choices of sea level changes reconstructions"
Based on comment 2 from the Editor, we rephrased this passage as:
*"In the case of a West Antarctic Ice Sheet collapse, which is the most probable scenario according to far-field sea level reconstructions, the range is further reduced to 6.7-8.2 m independently of the choices of external sea-level forcing and millennial-scale climate variability."*

**2.** Line 12: "choice of initial ice sheet configuration"
Added.

**3.** Line 13: Please reformulate " reproduce its recorded sea level high stand" into "to match the recorded global sea level high stand". This is because the sea level proxies record global signal and not only that of the Antarctic ice sheet and the big problem is to disentangle the individual signals.
That is a very good point, we have changed accordingly.

**4.** Lines 64-69: I would reformulate this paragraph first because atmospheric circulation over Antarctica is not homogeneous and this is what is evidenced by the different ice core records. Please include this statement in this paragraph. Also split the difference in temperature magnitude and the duration of the warmth. It will make this paragraph clearer.
We have reformulated the paragraph incorporating the mentioned suggestions. It now reads (L63):
*"Constraints are also scarce for the MIS11c climate, and its heterogeneity is reflected in the ice core records. Reconstructions from different ice cores located in East Antarctica (circles in Fig. 1) show different histories regarding the evolution of atmospheric surface temperature. For example, the Vostok ice core surface air temperature reconstruction (Petit et al., 1999; Bazin et al., 2013) reveals a weak temperature peak (about 1.6$^o$C above PI around 410 ka) compared to those of EPICA Dome C (EDC; over 2.7$^o$C above PI around 406 ka, Jouzel et al., 2007) and Dome Fuji (DF; 2.5$^o$C above PI around 407 ka, Uemura et al., 2018). The latter two ice-core records also present a peak-warming period of much longer duration (ca. 15 kyr compared to 7 kyr at Vostok)."*

**5.** Line 81: "sea level changes reconstructions"
We rephrased the expression as (L86): *"the choice of sea level reconstruction"*

**6.** Line 86: "terrain-following vertical layers"
Added "vertical" to the phrase.

**7.** Equation (5): I don't see the exponential in the formula. Please correct it.
We believe the reviewer might have been confused. There is no explicit exponential (i.e., the letter *e*) in the function, but the terms "1-GI(t)" and "GI(t)" are the exponents of $P_{PI}$ and $P_{LGM}$ respectively.

**8.** In sections 2.3.1 to 2.3.3: please refer to panels in Figure 2.
We added to the sentence before the last in section 2.3.1 (L177): "(orange and black dashed lines in Fig. 2b respectively)"

We already mention Fig. 2c in the last sentence of section 2.3.3 (L196).

**9.** Figure 4 caption: panel e) description is missing.
Thanks for spotting this, we have rephrased the part referring to the SLEN ensemble:
*"Panels d and e show floating and total ice volumes (in $10^6$ km$^3$), respectively, for the SLEN sea-level forcing reconstructions forced by EDC GI"*

**10.** Figure 5: is definitely too small. I would suggest to put two panels by raw in stead of all panels on the same raw. Also I would change the yellow color with orange or something more visible.
We changed it to a two-row figure as also suggested by Reviewer 1 in his comment 9. Regarding the color, we have changed the plotting order, making it significantly easier to see where the grounding lines differ.

**11.** Section 3.3: so actually SICOPOLIS grounding line scheme is highly insensitive to sea level changes… This is not the case of other ice sheet models. This is an important point also to be somehow discussed at the end of this paper in terms of uncertainties in those simulated volume and scenarios due to the model physics.
We had already mentioned our model insensitivity to sea-level forcing in the paragraph starting at line 360. We aimed to show that, compared to other external forcings, it played a minor role – which is similar in other models. Also, we acknowledge that this could be a limitation of the spatial resolution used (L360):
*"Despite differences in the model sensitivity to ocean temperature, our results support those of Tigchelaar et al. (2019) and Albrecht et al. (2020) regarding the minor role that variations in sea level play in driving ice-sheet retreat compared to other external forcings. Although the coarse treatment of the grounding lines could have had an influence on the seeming insensitivity of our experiments to sea-level uncertainties, other models of similar resolution which apply different sub-grid parameterisations to the grounding lines yield similar results (Tigchelaar et al., 2019, Sutter et al., 2019, Albrecht et al., 2020). Hence, while this caveat must be taken into consideration, it does not appear to have influenced our results dramatically."*

**12.** Figure 7 and figure 8: same as for Figure 5, perhaps better in two raws than all panel on a single raw. It is up to you.
We have made the same changes as in the response to comment 10 (i.e., Fig. 5).

**13.** Line 283: are you sure "height of MIS11c" is a correct English wording? Sounds a bit odd. Perhaps "peak of MIS11c" would be better? Not sure…just asking.
We believe it is correct, but agree that it is not standard. Thus, we have changed it to *"peak of MIS11c"*

**14.** Line 232 and 354-368: I think that you could simplify quite a lot the paragraph here. The main point is that LR04 is a stack of 57 globally distributed sediment cores with very few cores in the Southern Ocean…Thus LR04 represents a global averaged signal and not a local polar signal, in contrast with all the other ice core records…To me this is an important difference that also explains most of the timing and magnitude difference you see in all your simulations.

We reorganised the paragraph, precisely focusing on why LR04 misses the warming recorded by the ice cores (L375):

*"The LR04 reconstruction is composed of a stack of 57 globally-distributed ocean sediment cores (Lisiecki & Raymo, 2005), with a strong deficit over the Southern Ocean. In the Nordic Seas, paleoceanographic records indicate that the ocean was colder than present during MIS11 (Bauch et al., 2000; Kandiano et al., 2016; Doherty & Thibodeau, 2018). Colder ocean temperatures in the Northern Hemisphere explain why LR04 shows oxygen isotopic values similar to the Holocene during MIS11c (Lisiecki & Raymo, 2005) despite the geological evidence that there was a contribution to higher-than-Holocene sea levels from both Greenland and Antarctica (Scherer et al., 1998; Raymo & Mitrovica, 2012). Hence, the inclusion of many Northern Hemisphere records in the LR04 stack explains why it fails to capture the Antarctic warming during MIS11c seen in the ice cores, and the differences in timing compared to them. This also helps explain why the different criteria adopted for changing its scaling procedure had little effect on the results (Fig. 4b). A possible way of circumventing this problem could be to adopt a similar scaling approach to Sutter et al., (2019), who combined the LR04 stack and EDC ice-core temperature records, which, in their study, also led to WAIS collapse during MIS11c."*

**15.** From the supplementary CCSM3 tends to be colder than what is suggested by proxies: don't you think that it might anyway affect your spin-up and thus velocities and thus makes EAIS marine-based basin particularly insensitive to oceanic warming?

During the thermal spin-up we do not use the CCSM3 LGM climate. As stated in the methods (last paragraph of Sect. 2.1), we use the surface temperature from EDC as an anomaly to PI climate. The CCSM3 LGM climate is only used to produce the anomalies that are modulated by the GI during the 425-394 ka experiments.

**16.** Supplementary figure S14: Another interesting point this that comparing the dD and the derived dT° from your glacial index, it seems that if that glacial index would not overestimate the warmth at DF, simulations using DF would perhaps not lead to a WAIS collapse or perhaps delay it a lot. Just something to keep in mind. Here the fact the your glacial indices overestimate systematically the proxies results from the fact that LGM is too cold in the CCSM3 climate forcing, while PI is acceptable. This is a direct impact of your CCSM3 bias, despite the fact that you say in the supplementary that it would not affect your result. I think it does. So perhaps, just a sentence in the discussion of the main manuscript about this would be appreciated.

The reviewer offers two good points, which we address below.

First, we mention the cold bias of CCSM3, which does not appear to be fully addressed by the lapse-rate correction. We add after the first mention of Fig. S14 (L304):

*"[…] (Supplementary Fig. S14). This is most likely due to the cold bias in CCSM3, which persisted despite the lapse-rate correction applied. Nevertheless, Vostok's GI-reconstructed temperature peak matches the peak observed in DF […]"*

Second, we agree that if the GI-derived temperatures were not overestimated, the DF experiment would resemble the "extended Vostok-peak warmth" experiment, which further strengthens our  point that the

duration of warmer-than-present temperatures played a crucial role during MIS11c. We added the following to the discussion (L348):

[revised manuscript text omitted]

---

## Author Response (AR3)

Dear Editor,

We would like to thank you for your time in revising our submitted version of the manuscript. The points raised (presented below in black) were important to be clarified, and below we present (in green) how we addressed them. We have accepted all proposed suggestions, revised all calculations pointed out, and showed which passages were updated after the revisions (*in blue italics*). We have also updated the caption of fig. S15, which was indeed incomplete. Finally, we present a "latexdiff" version of the manuscript so that the changes made are visible.

We hope that you find our manuscript suitable for publishing in The Cryosphere.

Best regards,

Martim Mas e Braga

**MIS11 and MIS11c:** these terms appear to be used somewhat interchangeably, please clarify whether there is a difference between them, and if there is, then check that terms are used consistently throughout the text.

Our distinction between MIS11 and MIS11c was indeed not clear. We opted to change most mentions of MIS11 to MIS11c, because the statements were also valid for the substage. We only kept MIS11 where we make reference to studies which mention MIS11 itself, and not its substage (lines 56 and 386). We mention that MIS11 is the stage that encompasses MIS11c at line 56. For this same reason, the manuscript's title was changed to

*"Sensitivity of the Antarctic ice sheets to the warming of Marine Isotope Substage 11c"*

Which is more in line with how we introduce the substage both in the abstract and at the beginning of the introduction.

**Timing of the highstand:** it seems to be assumed that the MIS11c sea level highstand was at 405 ka. If this is the case, please state this clearly somewhere, referring to supporting evidence. If the highstand is not independently constrained to be at 405 ka, then do your results perhaps suggest that the lower bound for the AIS contribution to the highstand is 4 m – based on the Vostok scenario (fig. 6), where the sea level contribution is ~4 m at 410 ka?

We had assumed the highstand to be at 405 ka because both EDC and DF seem to be at their maximum during this point in time. The fact that this is not true for Vostok is indeed a good point, and thus we consider the contribution from Vostok at 410 ka instead. We further clarify this by slightly rewriting in line 294:

*"Given the observed spread, the three ensemble members constrain the range of potential sea level contributions from Antarctica during the MIS11c highstand to 4.0--8.2 m (minimum from Vostok at 410 ka, maximum from EDC at 405 ka). This range of 4.2 m essentially corresponds to whether the WAIS has collapsed or not during MIS11c."*

We also updated Fig. 11 to include the estimates for Vostok at 410, as opposed to 405 ka, thus generalising as "the highstand". We explain in the figure caption that it corresponds to 405 ka for EDC and DF, and 410 ka for Vostok. The interval was also updated throughout the entire text from 3.2-8.2 to

4.0-8.2 m s.l.e. (L8, 295, and 406). The individual contributions were also updated from 1.1-3.7 to 1.7-3.7 m s.l.e. for the EAIS (L413) and from 2.0-2.1 to 2.0-2.2 m s.l.e. for the WAIS (L415). Finally, we note that the EAIS contribution stated in L447 was also corrected from 2.4-3.7 to 2.3-3.7 m s.l.e.

**CCSM3 cold bias:** the statement on lines 304-305 requires further clarification. If I have understood the reviewer correctly, the CCSM3 LGM climate is too cold (even after correcting for the lapse rate), but the CCSM3 PI climate is relatively accurate. Due to the approach used to create the GI, this therefore leads to a cold bias for positive GI values and a warm bias for negative GI values (i.e. during an interglacial) – as shown in figure S14. Is this correct?

Yes, this is correct. We made it clearer in the text, by expanding the mentioned statement (L310):

*"This is most likely due to the LGM cold bias in CCSM3, which persisted despite the lapse-rate correction applied. Since PI temperatures do not have any strong bias, the LGM cold bias causes the GI reconstruction to yield colder temperatures during glacial times (GI > 0), and warmer temperatures during warmer-than-PI times (GI < 0)."*

Minor points

Line 8: 'contributed 3.2-8.2 m to…'
Done.

Line 10: delete 'further'
Done.

Lines 12-13: it is not clear how the climate signal is linked to global sea level, or what it means to 'match the recorded global sea level highstand'. Text could be tightened and perhaps linked more closely to information in the final sentence

We tried to make it clearer that the signal was needed so that the contribution expected from Antarctica was reproduced. We also tried to better link it to the following sentence (L12-15):

*"We found that the warmer regional climate signal captured by Antarctic ice cores during peak MIS11c is crucial to reproduce the contribution expected from Antarctica during the recorded global sea level highstand. This climate signal translates to a modest threshold of 0.4 °C oceanic warming at intermediate depths, which leads to a collapse of the West Antarctic Ice Sheet if sustained for at least 4 thousand years."*

Line 18: delete 'ka'
Done.

Line 51: make it clear that the Dry Valley moraines are interpreted to indicate local ice advance

We clarify it by slightly rephrasing the sentence (L51):

*"Counter-intuitively, the dating of onshore moraines in the Dry Valleys to MIS11c, indicating local ice advance, has been used to indirectly support regional ice sheet retreat (Swanger et al., 2017)."*

Line 68: 'apeak' – space missing
Added the space.

Line 79: To improve the structure of this paragraph, I suggest first stating that model results depend on forcings, boundary conditions, model parameters etc. You can then summarise which areas have previously been studied, before highlighting which aspects you will focus on.
As suggested, we added a sentence to introduce the paragraph (L80):

*"Ice-sheet model simulations depend on applied forcings, boundary conditions, and parameterisations for a wide range of processes. Such parameters control, for example, basal sliding, ice deformation, bedrock deformation, ice-shelf basal melting, and ice-shelf calving."*

Line 84: 'could help guiding' – check grammar
We corrected to the infinitive form.

Line 85: Awkward sentence, suggest, "We evaluate the impact of the following on AIS volume and extent during MIS11c: the choice of…
We thank the Editor for the suggestion, which we used. The sentence now reads (L88):

*"We evaluate the impact of the following on AIS volume and extent during MIS11c: the choice of proxy record (including their differences in signal intensity and structure), the choice of sea level reconstruction, and uncertainties in assumptions regarding the geometry of the AIS at the start of MIS11c."*

Figure 1: I do not see any red diamonds indicating sediment cores
This was a remnant of an old version of the manuscript, which included a reference to the sediment cores. We have removed it from the figure's caption:

*"Locations mentioned in the text are showcased, including the drilling sites of the ice cores used in this study (circles)."*

Line 96: this looks like eq. 9 in Bernales et al. (2017a), not eq. 1. Should $u_{ssta}$ be multiplied by w?
Thanks for spotting this, there was an error in the citation. The correct reference is Bernales et al. (2017b), where the equation is actually presented as Eq. (1). We corrected for this typo.

Line 127: your approach does not include all aspects of glacial isostatic adjustment (an important component is the spatial variation in sea surface height), suggest replacing with 'bed deformation'
We replaced "glacial isostatic adjustment" with the suggested term.

Line 142: 'When analysing the results, we ignore…'
We made the suggested change.

Line 144: Clarification needed because the EDC record was not used to force all the ensemble runs, e.g., it was not used to force all the CFEN experiments
We clarified that the EDC record is not used in all CFEN members by slightly rewriting the mentioned sentence (L146):

*"The EDC ice core was chosen for the thermal spin-up and as common forcing for all ensemble runs except for CFEN, where we test different core-derived climate signals (see below), because it spans the longest period among the three ice cores tested, while still providing a relatively high temporal resolution."*

Line 157: '…assess the impact of similarities and differences…'
We added "the impact of" as requested.

Table 2 caption: clarify that 'Age (ka)' relates to LGM reference values
We added this clarification to the caption:

*"Ice and sediment cores reference values used in Eq. (3), together with the age (in thousand years before present; ka) from which the LGM reference values were obtained."*

Line 187: delete 'Mean'. Also, given that your approach does not account for local gravitational perturbations to sea surface height, I suggest adding a sentence: "We approximate the sea level forcing applied at the boundaries of the ice sheet using global mean sea level reconstructions."
We added the suggested statement slightly rephrased, so it better fits the paragraph (L194):

*"For each ensemble member, the sea level forcing applied at the boundaries of the ice sheet is approximated to the global mean sea level of its respective sea level reconstruction."*

Lines 214-221: this text describes the initial ice sheet configurations (gmt1-gmt3) for the EDC case (shown in figure 3). Does it also hold for the cases when DF and Vostok forcing are used?
Yes, the configurations differ very little between experiments, and only in the magnitude of the differences relative to their respective control. We added this information to the caption of Figure 3, where it is mentioned that the EDC case is displayed:

*"Figure 3. (a-c) Three different starting ice sheet geometries at 420 ka for gmt1--3 using EDC forcing. The EDC CFEN member is used as "control". The same spatial pattern is seen for the DF and Vostok cases, and the averaged ice elevation difference between their respective geometries amounts to less than 50 m. Color scheme shows differences in surface elevation between each geometry and the control for 420 ka (d). Differences are only shown where the ice is grounded in both geometries, and coloured lines show the respective grounding lines in gmt1-3, also overlain in (d)."*

Lines 218 and 219: '...than the control…'
Corrected.

Table 4: reference to $\delta X_{Hol}$ is perhaps left over from an earlier version of the manuscript?
Correct, we removed it.

Line 263: you state above (line 258) that using the LR04 average values gives a 3.4% smaller ice sheet at 402 ka, and here you state that using the EDC average value gives a 2.3% larger ice sheet at 402 ka. However, in figure 4b, the orange solid/dashed lines are much closer to each other at 402 ka than the black solid/dashed lines - please check calculations
We double checked the calculations, excluding any rounding in the numbers, and there is indeed a larger difference. The difference between $LR04_{LGMavg}$ and LR04 is indeed larger than previously reported ($1.1 * 10^6$ km$^3$), amounting to 4.2 % of LR04 at 402 ka. As for the difference between EDC and $EDC_{LGMavg}$, it is smaller than previously reported ($0.3 * 10^6$ km$^3$), and amounts to 1.2 % of EDC's volume at 402 ka. We corrected the numbers in the manuscript (L263 and L268).

Line 271: 'It directly reflects their effect' – references to 'it' and 'their' are ambiguous
We clarified it by replacing 'it' and 'their' (L277):

*"Thus, floating ice volume directly reflects the sea level forcing effect on the flotation of ice, and consequently on the grounding line position."*

Line 284: 'different initial geometries'
Added 'initial' as suggested.

Line 285: 'The latter two…' – check, I think it is Totten and Dibble that are thicker, with Cook thinner
That is absolutely correct. We corrected the name order in the text accordingly, to Cook, Totten, and Dibble.

Line 318: 'the former two' – not clear what this refers to
We clarified to which they refer by rewriting the sentence (L326):

*"Nevertheless, they show limited retreat compared to the aforementioned WAIS ice shelves."*

Line 323: 'the different ice-sheet configurations' – make it clear that you are talking about model runs forced by the same ice core record, but with different initial ice sheet configurations
We added a clarification to the mentioned sentence (L330):

*"This observed tipping point at 412 ka also explains why the different initial ice-sheet configurations under a common forcing follow..."*

Line 340: clarify that the values relate to ocean temperatures
We clarified it in the sentence, just before mentioning the temperature values (L348):

"The Vostok-based simulations (Figs. 10e-h) show that there is indeed a threshold in ocean temperatures, which is of approximately 0.45 °C..."

Line 356: 'WAIS collapse was triggered' – more caution needed in the language used, it is not proven that WAIS collapsed during the LIG
We slightly rephrased the sentence, which now reads (L365):

*"In other interglacials, such as the LIG, the shorter duration but higher intensity of ocean warming compared to MIS11c could have triggered WAIS collapse (Dutton et al., 2015, Turney et al., 2020) ..."*

Line 370: 'ested' -> 'tested'
Corrected

Figure 10 caption: (b,e) -> (b,f)
Corrected

Line 391: 'when comparing their results' – check the logic in this sentence
We slightly rephrased the sentence (L399):

*"As for West Antarctica, far-field sea level reconstructions suggest that a WAIS collapse was the most probable scenario (Raymo & Mitrovica, 2012, Chen et al., 2014) when comparing global highstand estimates with the probable contribution from the GIS."*

Lines 397 and 399: suggest 'interval' -> 'range'
We made the suggested changes.

Line 398: 'ice core experiments'
Done.

[revised manuscript text omitted]